# Iron sulfide-catalyzed gaseous CO$_2$ reduction and prebiotic carbon fixation in terrestrial hot springs

Jingbo Nan [1,13], Shunqin Luo [2,13] ✉, Quoc Phuong Tran [3,4], Albert C. Fahrenbach[3,4,5], Wen-Ning Lu[2,6,7], Yingjie Hu[8], Zongjun Yin [1], Jinhua Ye [2,9,10] ✉ & Martin J. Van Kranendonk [4,11,12]

Understanding abiotic carbon fixation provides insights into early Earth's carbon cycles and life's emergence in terrestrial hot springs, where iron sulfide (FeS), similar to cofactors in metabolic enzymes, may catalyze prebiotic synthesis. However, the role of FeS-mediated carbon fixation in such conditions remains underexplored. Here, we investigate the catalytic behaviors of FeS (pure and doped with Ti, Ni, Mn, and Co), which are capable of H$_2$-driven CO$_2$ reduction to methanol under simulated hot spring vapor-zone conditions, using an anaerobic flow chamber connected to a gas chromatograph. Specifically, Mn-doped FeS increases methanol production five-fold at 120 °C, with UV–visible light (300–720 nm) and UV-enhanced light (200–600 nm) further increasing this activity. Operando and theoretical investigations indicate the mechanism involves a reverse water-gas shift with CO as an intermediate. These findings highlight the potential of FeS-catalyzed carbon fixation in early Earth's terrestrial hot springs, effective with or without UV light.

Transition-metal sulfides are known for catalyzing a wide range of reactions[1]. Iron sulfides, in particular, have been proposed to play an important role in the origin of life[2–6]. As capable electron conductors, they are thought to have performed critical protometabolic functions in prebiotic chemistry. Specifically, the cubane structure of greigite (Fe$_3$S$_4$), which forms through the oxidation/sulfidation of mackinawite (FeS), is often linked to the [4Fe4S] clusters found in the active sites of many redox enzymes and electron carriers such as carbon monoxide dehydrogenase/acetyl-CoA synthase (CODH/ACS), which participate in the ancient acetyl-CoA pathway[7], and ferredoxins[5,8,9]. Computational and phylogenetic studies[10,11] also suggests that the last universal common ancestor (LUCA) relied on iron–sulfur cluster catalysis.

Studies on iron sulfides in prebiotic chemistry have predominantly focused on submarine alkaline hydrothermal vents, which are considered potential cradles for life's emergence[5,6,12–15]. The alkaline vent theory (AVT) suggests that CO$_2$ reduction by hydrothermal H$_2$ could have served as a starting point for carbon fixation pathways at the origin of metabolism[4,16–18]. In this scenario, the interaction of

[1]State Key Laboratory of Palaeobiology and Stratigraphy, Nanjing Institute of Geology and Palaeontology, Chinese Academy of Sciences, 210008 Nanjing, China. [2]International Center for Materials Nanoarchitectonics (WPI-MANA), National Institute for Materials Science (NIMS), 1-1 Namiki, Tsukuba, Ibaraki 305-0044, Japan. [3]School of Chemistry, University of New South Wales, Sydney, NSW 2052, Australia. [4]Australian Centre for Astrobiology, University of New South Wales, Sydney, NSW 2052, Australia. [5]UNSW RNA Institute, University of New South Wales, Sydney, NSW 2052, Australia. [6]National Key Laboratory of Uranium Resource Exploration-Mining and Nuclear Remote Sensing, East China University of Technology, 330013 Nanchang, China. [7]State Key Laboratory of Nuclear Resources and Environment, East China University of Technology, 330013 Nanchang, China. [8]Nanjing Key Laboratory of Advanced Functional Materials, Nanjing Xiaozhuang University, 211171 Nanjing, China. [9]Graduate School of Chemical Sciences and Engineering, Hokkaido University, Sapporo, Hokkaido 060-0814, Japan. [10]TJU-NIMS International Collaboration Laboratory, School of Materials Science and Engineering, Tianjin University, 300072 Tianjin, China. [11]School of Biological, Earth, and Environmental Sciences, University of New South Wales, Sydney, NSW 2052, Australia. [12]Present address: School of Earth and Planetary Sciences, Curtin University, Bentley 6845, Western Australia. [13]These authors contributed equally: Jingbo Nan, Shunqin Luo. ✉e-mail: luoshunqin1116@163.com; jinhua.ye@nims.go.jp

seawater and Fe(II)-silicates within the oceanic lithosphere of the early Earth resulted in alkaline hydrothermal fluids that were enriched in dissolved $H_2$ and sulfides[19]. Specifically, the interaction between alkaline fluids and the acidic prebiotic seawater is proposed to have led to the formation of carbonate chimneys containing iron sulfide precipitates[5,20]. When $H_2$-bearing fluids came into contact with $CO_2$-bearing seawater, catalytically active iron sulfides would facilitate $CO_2$ reduction. Moreover, the proton gradient, as a result of the physical separation between high and low-pH fluids by iron sulfide-bearing inorganic membranes, has been proposed to yield the electrochemical energy necessary to catalyze carbon fixation[4,21]. As such, this far-from-equilibrium nature of hydrothermal vents has exemplified their potential significance in the origin of metabolism[22].

An alternative hypothesis posits that life may have originated in terrestrial hot springs, where the energy from geothermal activity and sunlight irradiation may have facilitated $CO_2$ reduction[23–25]. In this scenario, $CO_2$ was a dominant gas in the early Earth atmosphere[26–28], a fraction of which would have dissolved in hot spring fluids as dissolved $CO_2$ and (bi)carbonate. $H_2$ derived from fluid–rock interactions (such as serpentinization) could have been present as free gas and dissolved in groundwater[29–31]. Importantly, terrestrial hot springs also contain many of the dissolved ions found in deep-sea vents, such as iron, sulfide, and silica[23,29,32]. Yet, despite these similarities, the potential for FeS-catalyzed carbon fixation in prebiotic terrestrial hot spring scenarios and related mechanisms remains largely unexplored.

Here, we investigate gaseous $CO_2$ reduction catalyzed by iron sulfides under simulated terrestrial hot spring conditions. Pure FeS and FeS doped with various metals, Fe(M)S, proposed to have been commonly present in prebiotic hot spring systems[23,32,33] (e.g., Mn, Ni, Ti, and Co), are synthesized and tested. Kinetic analysis using gas chromatography equipped with a flame ionization detector (GC-FID) reveals that the FeS precipitates serve as catalysts in the $H_2$-dependent reduction of $CO_2$ to methanol ($CH_3OH$). Exposure to UV−visible light (300–720 nm) and UV-enhanced light (200–600 nm) promotes $CH_3OH$ production. Pure FeS and Mn-doped FeS demonstrate the most encouraging catalytic behaviors and their mechanisms are further investigated via experimental activation energy measurements, in situ diffuse reflectance infrared Fourier transform spectroscopy (DRIFTS), and quantum mechanical modeling. Although there is uncertainty regarding how persistent or temporary conditions, such as $H_2$, $CO_2$, heat, and light, were in early hot spring environments, if conditions similar to those tested in our experiments did occur, they would likely have driven the gaseous $CO_2$ reduction forward. As such, the comparison of this abiotic carbon fixation mechanism with the extant biotic acetyl-CoA pathway provides an additional empirical lens for critically assessing the idea that the origin of ancient carbon-fixation metabolisms may have direct roots in prebiotic chemistry.

## Results

### Synthesis and structural characterization of FeS

A series of nanoparticulate forms of FeS incorporating Mn, Ni, Ti, and Co were developed in order to investigate the hypothesis that $CO_2$ reduction may have occurred in terrestrial hot springs on early Earth. These FeS minerals were synthesized by simply mixing $Na_2S$ and $FeCl_2$ solutions under an oxygen-free environment (see the "Methods" section for details). The FeS precipitates were washed and suspended in $N_2$-purged MilliQ water. The suspensions were then mixed with pretreated $SiO_2$ powder that served as an inert support while promoting the dispersion of the FeS samples. The combined mixture was left to dry at 25 °C for 48 h.

Transmission electron microscopy (TEM) confirmed the formation of Mn-doped FeS as irregular or plate-like aggregates (Fig. 1a), composed of nanometer-sized small crystals. The size of these plate-like nanocrystals ranged from several to tens of nanometers; however, distinguishing individual crystals within the aggregates remains

challenging. High-resolution TEM (HR-TEM) analysis further identified lattice fringes of 5.0, 3.1, and 2.3 Å, which correspond to the (001), (101), and (111) planes of mackinawite (FeS), respectively (Fig. 1b, c). This observation suggests that the introduction of a minor amount of Mn does not significantly alter the crystalline structure of pure FeS, likely as a consequence of the similar atomic radii of Fe and Mn. The incorporation of Mn is also supported by energy dispersive X-ray (EDX) spectroscopy mapping, showing a nearly homogeneous distribution of Mn in the FeS (Fig. 1d–g, Supplementary Fig. S1). Ultraviolet−visible diffuse reflectance spectroscopy (UV−vis DRS), which measured the light absorption properties of pure FeS and Mn-doped FeS (Supplementary Fig. S2), revealed that both samples exhibit similar semiconducting features characterized by a band gap of approximately 2.51 and 2.47 eV (~500 nm), respectively. This slight decrease in the band gap observed upon introducing Mn into FeS is likely a result of Mn introducing impurity states within the band structure, which facilitate electron transitions at lower energy levels and thereby reduce the band gap[34,35]. The X-ray diffraction (XRD) pattern of FeS featured a short-range mackinawite ordering, manifesting an average crystalline size of 12.2 nm according to the Scherrer equation. Introducing Mn into FeS slightly increased the crystallinity, with the appearance of MnS in the Mn-doped FeS solid solution mixture (Supplementary Fig. S3 and Table S1).

### Screening the catalytic activity of (doped) FeS in the reduction of $CO_2$ by $H_2$

The activities of the different iron sulfides were investigated (Fig. 2a) under ambient pressure using a custom-built anaerobic open chamber with a temperature-controlled reaction bed (Supplementary Fig. S4). Different lamps were employed to simulate potential spectral ranges on the early Earth's surface. Constraints on the flux of UV light on early Earth have been estimated based on spectral output from a modeled young Sun[36]. Given that water vapor and $CO_2$ would have been dominant gases in the early Earth atmosphere, the shortest wavelengths able to reach the early Earth's surface start at ~200 nm[36]. This spectral range was simulated using a Hg−Xe lamp (200–600 nm, Supplementary Fig. S5), which is referred to as "UV-enhanced" light henceforth. However, various factors can also attenuate the UV flux in the 200–300 nm region, such as the presence of $SO_2$ and $H_2S$, which absorb lower UV wavelengths[29]. While global concentrations of these volcanic gases are calculated to be low due to photolysis and reactions with oxidants[36], their local concentrations surrounding terrestrial hot springs may have been high enough to block 200–300 nm light, especially during periods of high volcanic activity. This spectral range, referred to as "UV−visible" light, is simulated using an Xe lamp (300–720 nm, Supplementary Fig. S5a).

Prior to the experiment, 10 mg of the FeS−$SiO_2$ mixture was uniformly distributed on the reaction bed. Afterwards, $H_2$ (15 mL min⁻¹) and $CO_2$ (5 mL min⁻¹) gases ($H_2$/$CO_2$ = 15/5) were introduced into the chamber at a constant flow rate through the reaction bed and out to an inline GC-FID for quantitative analysis.

Under 120 °C and ambient pressure, all the tested iron sulfides, including pure FeS and FeS doped with different metals, exhibited the capability to catalyze $CO_2$ reduction to $CH_3OH$ (Fig. 2b and Supplementary Fig. S6). Lower catalytic performance upon the introduction of Ni, Co, and Ti into FeS was observed. This result could be attributed to the fact that the production of $CH_3OH$ as the final product of $CO_2$ hydrogenation requires optimal adsorption and desorption of reactants, critical intermediates, and products. In these experiments, the production of $CH_4$ was observed as one of the products when using Ni-doped FeS (Supplementary Table S2), suggesting that methoxy intermediates might be further hydrogenated to $CH_4$ before desorption[37], thereby reducing the yield of $CH_3OH$. Co-based materials are generally recognized as efficient catalysts for the Fischer-Tropsch process due to their strong capacity for the adsorption and partial dissociation of

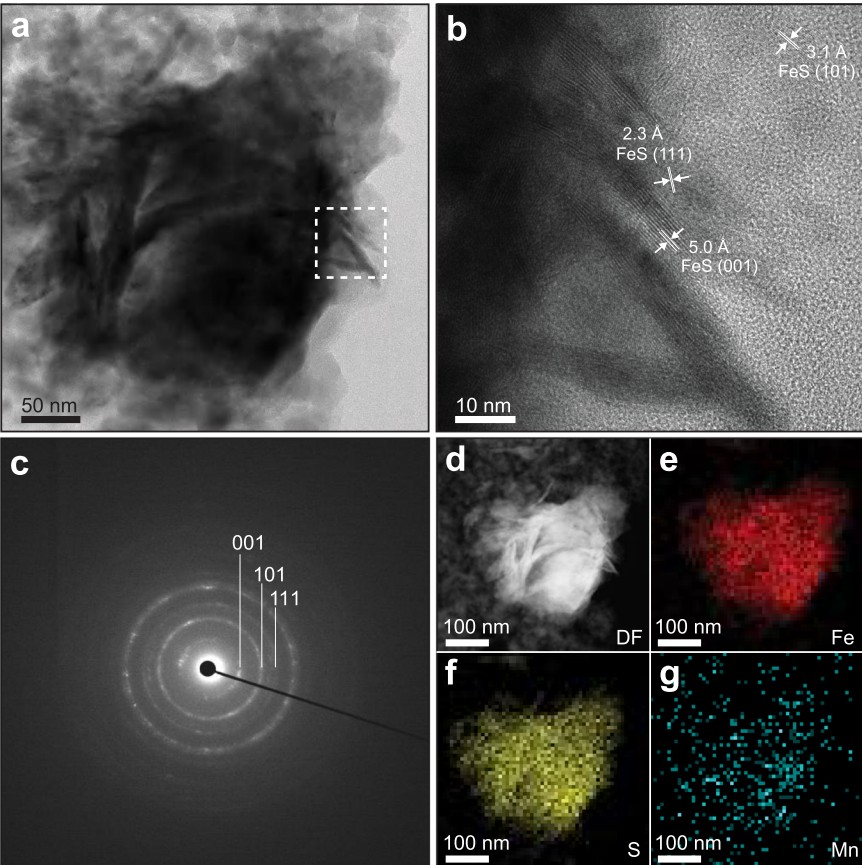

**Fig. 1 | Characterization of the Mn-doped FeS catalyst. a** Bright-field scanning transmission electron microscopy (STEM) image showing the irregular or plate-like crystalline structure of the resultant FeS nanoparticles. **b** High-resolution TEM (HR-TEM) (enlarged view of area boxed in an image showing lattice fringes of FeS nanoparticles). **c** The selected area electron diffraction (SAED) pattern showing diffraction rings consistent with mackinawite. **d**–**g** STEM image and corresponding energy-dispersive X-ray (EDX) spectroscopy mappings for iron (red), sulfur (yellow), and manganese (blue). DF dark field.

carbon species. Therefore, introducing Co into FeS may hinder the desorption of $CH_3OH$ due to the strong interaction between Co-active sites and $CH_3OH$[38]. A similar phenomenon may also occur in the case of Ti-doping, where Ti sites could facilitate the spontaneous dissociation of $CH_3OH$ through the formation of Ti–$OCH_3$, leading to lower $CH_3OH$ production[39]. These unique characteristics contribute to lower production and release of $CH_3OH$ in Ni-, Co-, and Ti-doped FeS compared to pure FeS. The catalytic activity towards $CH_3OH$ production was further enhanced by UV−visible light irradiation, the extent of which differed for each metal-doped iron sulfide. Notably, Mn-doped FeS exhibited the highest catalytic activity under both light and dark conditions, surpassing that of pure FeS (Fig. 2b and Supplementary Fig. S7). All the experiments included a series of control blank tests specifically designed to confirm the absence of organic contamination throughout the experimental procedures (see the "Methods" section, Supplementary Figs. S6 and S7).

To evaluate the robustness of $CH_3OH$ production under different $H_2/CO_2$ ratios, we also conducted experiments at lower $H_2/CO_2$ ratios of 5/15 and 10/10 on Mn-doped FeS. The results show that an increase in the $H_2/CO_2$ ratio leads to higher production rates of $CH_3OH$ (Supplementary Fig. S8a). Even at the lowest ratio (5/15), however, considerable production of $CH_3OH$ (up to 2.6 $\mu mol\,g^{-1}\,min^{-1}$) can be observed.

**Catalysis of $CO_2$ fixation by pure and Mn-doped FeS in a simulated hot spring vapor zone**
Subsequent experiments focused on pure FeS and Mn-doped FeS owing to their high catalytic activities. The production of methanol

using these catalysts was examined systematically under varying temperatures and irradiation conditions to understand their influence on the FeS-catalyzed reduction of $CO_2$. As shown in Fig. 2c, elevating the temperature from 80 to 120 °C resulted in an increased production rate of $CH_3OH$ for both pure and Mn-doped FeS samples. Similarly, UV−visible irradiation was also observed to promote $CH_3OH$ production in these iron sulfide samples. Mn-doped FeS consistently demonstrated superior catalytic activity in comparison to pure FeS, an observation that suggests that the introduction of Mn facilitates $CO_2$ hydrogenation.

Further experiments employing various Mn percentages, specifically 0.1%, 1%, and 10%, show observable improvement of FeS catalytic activity even at lower Mn-doping percentages (Supplementary Fig. S8b). The results indicate that in both dark and UV−visible conditions, $CH_3OH$ yield increases with higher Mn percentages. In contrast, experiments with pure MnS showed the lowest activity under both dark and light conditions, with only slight enhancement from light irradiation. MnS has limited capacity for $CO_2$ adsorption and activation, hindering its catalytic effectiveness[40]. Its wide band gap (~3.2 eV) makes it primarily responsive to UV light[41], which can degrade the as-formed $CH_3OH$ products. Conversely, pure and Mn-doped FeS respond to both UV and visible light, resulting in more efficient $CO_2$-to-$CH_3OH$ conversion.

To further explore the feasibility of FeS-catalyzed carbon fixation in terrestrial hot springs, the effect of water vapor and UV-enhanced exposure using a Hg−Xe lamp (with emission of $200 < \lambda < 600$ nm) on $CH_3OH$ production catalyzed by Mn-doped FeS was investigated (Fig. 2d and Supplementary Fig. S8c). The results show that the

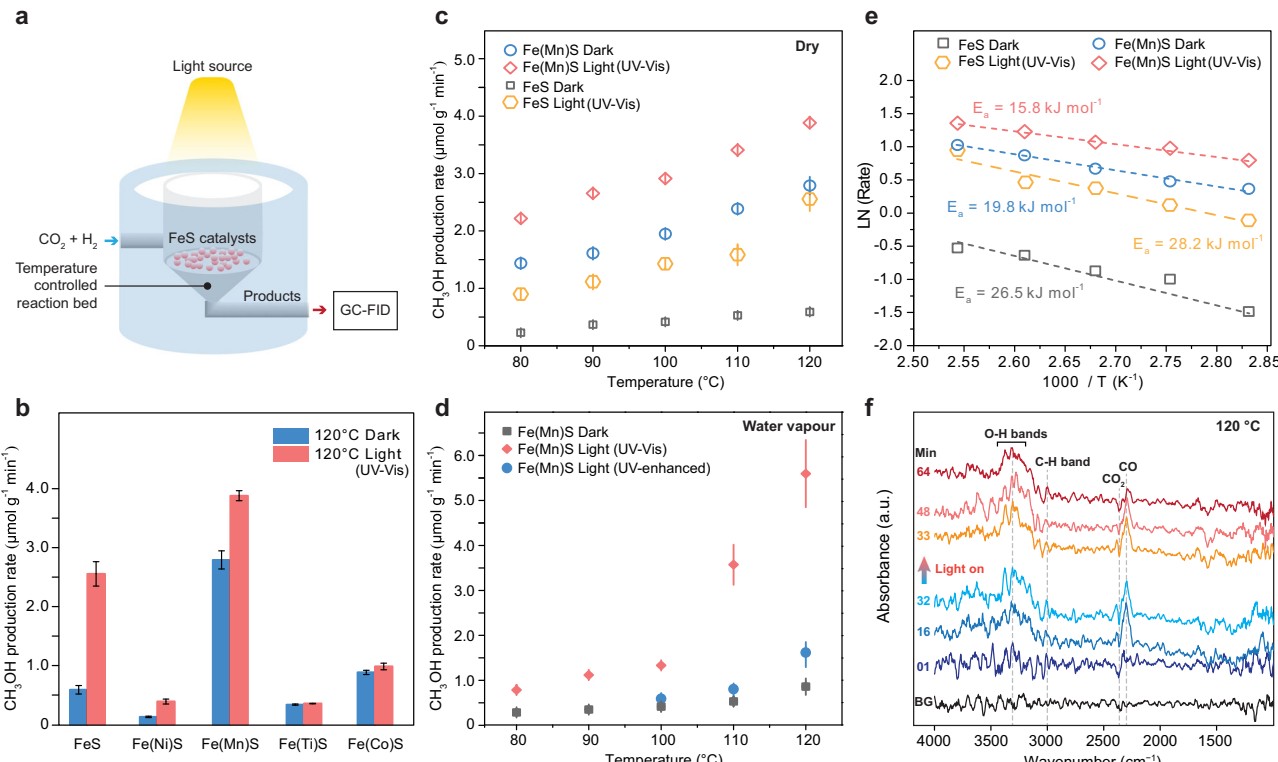

**Fig. 2 | CO₂ reduction with H₂ catalyzed by metal-doped FeS. a** Schematic representation of the reaction cell used for the light-mediated thermocatalytic reaction. **b** Effect of metal doping on FeS catalysis under dark and UV−visible light (300−720 nm) conditions. **c** Temperature-dependent experiments of CH₃OH production from CO₂ and H₂ over pure FeS and Mn-doped FeS under dark and UV−visible light conditions. **d** Temperature-dependent experiments of CH₃OH production in the presence of water vapor over Mn-doped FeS under dark conditions and with the irradiation of UV−visible light or UV-enhanced light (200−600 nm). The error bars in **b**−**d** were generated from a triplicate set of measurements. **e** The apparent activation energies ($E_a$) over FeS and Mn-doped FeS catalysis with and without UV−visible light irradiation. **f** In situ diffuse reflectance infrared Fourier transform spectra (DRIFTS) of Mn-doped FeS under dark and UV−visible light conditions at 120 °C. Unless otherwise specified, all metal-doped FeS refers to FeS doped with 10% metal.

presence of water vapor led to a ~50% reduction in CH₃OH production at lower temperatures (80–100 °C), while at higher temperatures (100–120 °C), the same water vapor content enhanced the yield by ~50%. These observations indicate that water vapor only inhibits CH₃OH synthesis at temperatures below the boiling point of H₂O. It is hypothesized that at higher temperatures (above 100 °C), water vapor may actively engage in CO₂ hydrogenation by contributing protons. This process is likely facilitated by increased kinetic energy at higher temperatures, enhancing the interaction between water vapor and CO₂ and resulting in a more efficient hydrogenation process. UV-enhanced irradiation was observed to have a detrimental effect on CH₃OH production. Methanol production upon UV-enhanced irradiation was below the detection limit at 80–90 °C but was observed for 100–120 °C (Fig. 2d). This decrease in methanol yield may be a consequence of the higher energy UV light aiding in the decomposition of adsorbed CH₃OH or other intermediates formed on the surface of the catalysts before completion of the overall reaction cycle. Nevertheless, these findings suggest that H₂-driven CO₂ reduction catalyzed by FeS could have occurred over a range of UV flux scenarios and temperatures conceivable for a terrestrial hot spring setting on early Earth.

### Mechanism of gaseous CO₂ reduction over FeS

Additional mechanistic understanding of the effects of Mn-doping and UV−visible light irradiation on CH₃OH production was obtained through the determination of apparent activation energies for CO₂ reduction by fitting observed reaction rates ($\mu mol\,g^{-1}\,min^{-1}$) to Arrhenius plots (Fig. 2e, Methods). Notably, the apparent activation energy for that of FeS was lowered from 26.5 to 19.8 kJ mol⁻¹ by Mn doping, an observation suggesting that Mn might serve as an active site to facilitate the rate-determining step of CO₂ hydrogenation. The apparent activation energy of Mn-doped FeS was further reduced to 15.8 kJ mol⁻¹ by UV−visible light irradiation (Xe lamp), culminating in a total 40% decrease compared to that for pure FeS under dark conditions. This significant decrease in activation energy is an indication that light irradiation promotes surface reactions likely by exciting FeS, which generates photo-induced charge carriers that further enhance reaction at the Mn sites. No substantial change in apparent activation energy was observed when pure FeS was exposed to the same light source, a result that is consistent with the absence of active sites that can deliver the photo-induced charge carriers to facilitate the rate-determining step of the surface reaction.

To understand the CH₃OH synthesis pathway, in situ DRIFTS spectra were recorded at 120 °C under both dark and UV−visible irradiation conditions (Fig. 2f). The background spectrum was initially recorded at 120 °C after the adsorption−desorption equilibrium between the reactants and catalysts had been reached, followed by sealing the reaction chamber. The production of CH₃OH through CO₂ reduction was previously shown to involve the formation of *CO, *HCHO, or *HCOOH as intermediates[42,43] (asterisk '*' denoting adsorbed species).

Throughout the in situ DRIFTS measurements conducted, a downward peak of CO₂ at 2357 cm⁻¹ was observed, indicating the consumption of CO₂ during the reaction, while an upward peak at ~2298 cm⁻¹ demonstrated the formation of *CO. This result indicates that *CO is likely a critical intermediate during the reaction and suggests that the reaction proceeds through the reverse water gas shift (RWGS, CO₂ + H₂ → CO + H₂O) pathway followed by the hydrogenation of *CO to yield CH₃OH. As the reaction progressed, a gradual decrease

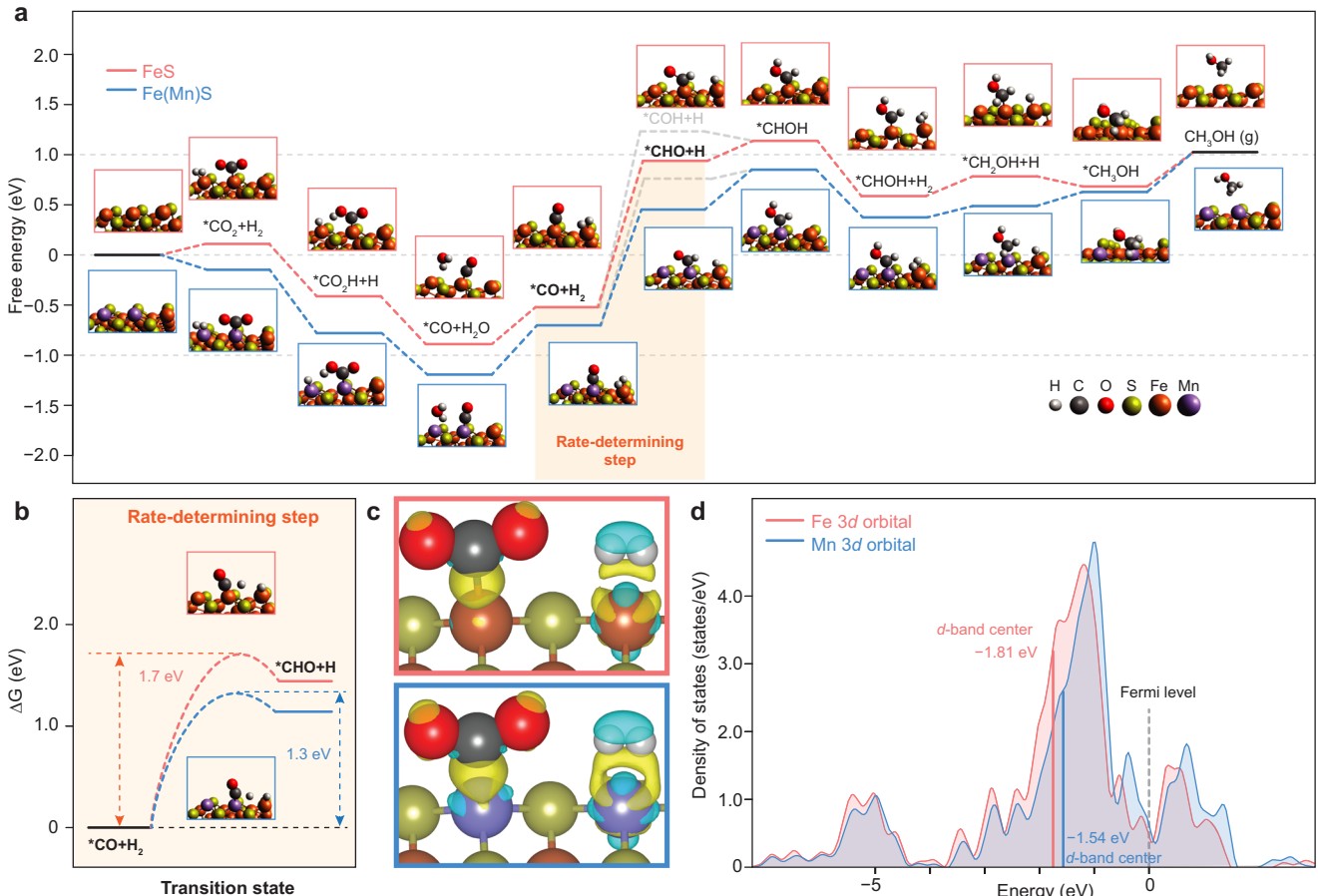

**Fig. 3 | Density functional theory (DFT) calculations of CO₂ hydrogenation on the FeS and Mn-doped FeS (100) surfaces. a** Gibbs free energy diagram of the reaction steps and corresponding optimized structures. **b** Transition states of the rate-determining step. **c** Calculated charge density differences (CDD) between pure FeS (upper) and Mn-doped FeS (lower) upon adsorption of CO₂ and H₂. Yellow represents electron accumulation, and cyan denotes electron depletion. **d** Calculated density of states (DOS) of Fe and Mn sites in FeS.

in the intensity of *CO was recorded, suggesting further hydrogenation/consumption of *CO to the final products. Consequently, noticeable upward broad bands in the 3200–3500 cm⁻¹ range were detected, which are attributed to the O−H stretching of CH₃OH and H₂O. Relatively weak peaks assigned to C−H vibrational bands around 3000 cm⁻¹ further confirmed the formation of CH₃OH upon successive hydrogenation of *CO. The intensity of O−H stretching bands (3200–3500 cm⁻¹) was seen to increase over time under dark conditions, and this increase was found to be further accelerated after UV–visible irradiation. The absence of additional absorption peaks under irradiation provides evidence that the photo-induced charge carriers do not significantly alter the reaction pathway.

Density functional theory (DFT) quantum mechanical calculations were employed to investigate the role of pure and Mn-doped FeS during CO₂ hydrogenation at an atomic scale. Pioneering studies have demonstrated that the surface of iron sulfide compounds, such as FeS, exhibits a unique capacity for CO₂ adsorption and activation[44,45], while Fe₃S₄ has been shown to catalyze the conversion of CO₂ into small organic molecules, such as formic acid and methanol[46]. In order to gain a comprehensive understanding of the thermodynamics of the whole reaction process, Gibbs free energies for each elementary step were calculated (Fig. 3a). The main intermediates were determined to be *CO₂, *CO₂H, *CO, *CHO (*COH), *CHOH, *CH₂OH, and *CH₃OH. The conversion of CO₂ to *CO₂H, *CO, and H₂O is exothermic on both the pure and Mn-doped FeS surfaces. Subsequent hydrogenation of *CO to *CHO or *COH exhibited higher Gibbs free energies than the starting materials. Even though both the formation of *COH via *CO

hydrogenation (involving O−H bond formation) and the generation of *CHO (involving C−H bond formation) are endothermic processes, the pathway leading to *CHO is more thermodynamically favorable. Therefore, the formation of *CHO through *CO is likely the rate-determining step (RDS). Further hydrogenation of *CHO generates CH₃OH as the final product.

Detailed calculations show that the overall CO₂ hydrogenation process was more feasible on Mn-doped FeS than on pure FeS (Fig. 3a). On pure FeS, CO₂, and H₂ were found to be adsorbed on the Fe sites with a positive energy barrier of 0.11 eV, indicating that FeS has a relatively inert surface for adsorbing and activating CO₂ molecules. Upon substituting several surface Fe atoms with Mn, CO₂ and H₂ adsorption was significantly enhanced, a feature which could help explain the facilitation of the subsequent conversion of CO₂ into intermediates such as *CO₂H and *CO. Hydrogenation of *CO to *CHO (the RDS) exhibited much lower energies on Mn-doped FeS than that of the pure FeS surface. The optimized transition states of these RDSs demonstrate significantly lower energy barriers on Mn-doped FeS compared to pure FeS (1.3 vs. 1.7 eV), revealing a pivotal role of Mn active sites in assisting C−H bond formation (Fig. 3b). Overall, the predicted reaction dynamics are consistent with the apparent activation energies and support the mechanism that Mn active sites facilitate both CO₂ adsorption and the RDS of *CO hydrogenation.

To understand how Mn facilitates the adsorption of CO₂ and H₂, the charge density differences (CDDs) of pure and Mn-doped FeS after adsorbing CO₂ and H₂ were calculated based on DFT. Figure 3c highlights the sites where electron accumulation and depletion occur upon

adsorption and illustrates the orbital interactions between $CO_2/H_2$ and the catalyst surfaces[47,48]. A pronounced difference in the CDD between pure FeS and Mn-doped FeS is observed. Specifically, when $CO_2$ and $H_2$ approach the catalyst, an accumulation of electron density is seen around the Fe and Mn sites. The substitution of Mn in the FeS lattice appears to create a site with enhanced electronic density compared to pure FeS. This increased electron density at the Mn site may facilitate greater adsorption of $CO_2$ and $H_2$ as a consequence of stronger electron donation and could be the reason for their predicted lower energy barriers for adsorption on the Mn-doped FeS surface.

Density of state (DOS) calculations were further conducted to elucidate the strength of adsorption at the Fe and Mn sites (Fig. 3d). By analyzing the Kohn–Sham orbitals for the new bands near the Fermi level (the highest energy level that electrons can occupy at absolute zero temperature), it was found that these states are mainly located around the isolated Fe and Mn sites. The doping of Mn is observed to result in an appreciable upshift of the $d$-band center from −1.81 to −1.54 eV. It is a well-established principle that an upshifted $d$-band center tends to elevate more antibonding states above the Fermi level, thereby leading to a reduction in occupation and, consequently, an enhanced strength in adsorbate bonding[49]. The integration of Mn atoms shifts the $d$-band center closer to the Fermi level, indicating a potential augmentation in $CO_2$ and $H_2$ molecules bonding to the surface. The differential charge densities over the $CO_2$-adsorbed structural models display a pronounced transfer of electrons to the near-surface region of Mn-doped FeS, which highlights the altered electronic interactions facilitated by Mn doping.

## Discussion

This study demonstrates that FeS nanoparticles can catalyze $H_2$-dependent gaseous $CO_2$ fixation into $CH_3OH$ under moderate temperatures (80–120 °C) and ambient pressures in a neutral atmosphere, conditions that are plausible in terrestrial hot spring settings on early Earth. Specifically, GC-FID was used to quantify the catalytic activity of pure FeS as well as FeS doped with metals found in hot spring fluids (e.g., Ti, Mn, Co, and Ni). Carbon fixation activity was the highest in Mn-doped FeS, followed by pure FeS. Furthermore, it was found that broad-range light irradiation (200–720 nm) increased the catalytic activity of both these FeS materials while also lowering their activation energies. Notably, the introduction of water vapor does not entirely inhibit Mn-doped FeS activity and actually increases it at temperatures above 100 °C. Although the UV-enhanced light source was observed to be detrimental to $CH_3OH$ production, significant amounts of $CH_3OH$ were still produced at temperatures ≥100 °C under these harsher irradiation conditions.

In early Earth hot spring scenarios (Fig. 4), rising groundwater mixes dissolved sulfide with additional Fe, Ti, Mn, Co, and Ni[23,29,33] to form iron sulfide particulates that deposit on the edges of the hot springs. These iron sulfide deposits are then exposed to $H_2$ gas emitted from the ground[31] (e.g., via serpentinization, magma degassing, or other processes), as well as $CO_2$[26–28]. Therefore, our study demonstrates that FeS-catalyzed carbon fixation could have occurred at such terrestrial hot spring environments, albeit via a different mechanism than in deep-sea alkaline vents. In addition to offering unique gas-solid interfaces like those studied herein, terrestrial hot spring scenarios feature exposure to sunlight, an energy source absent in deep-sea vent environments. While UV light might be detrimental to the production and accumulation of organics in terrestrial environments, it is also thought to be critical for the prebiotic synthesis and selection of many molecules considered important for prebiotic chemistry[50–54]. Moreover, local concentrations of $SO_2$ and $H_2S$ surrounding terrestrial hot springs may have been high enough to block lower wavelengths (200–300 nm)[29], especially during periods of high volcanic activity. Although light is not necessary for the conversion of $CO_2$ to $CH_3OH$, as

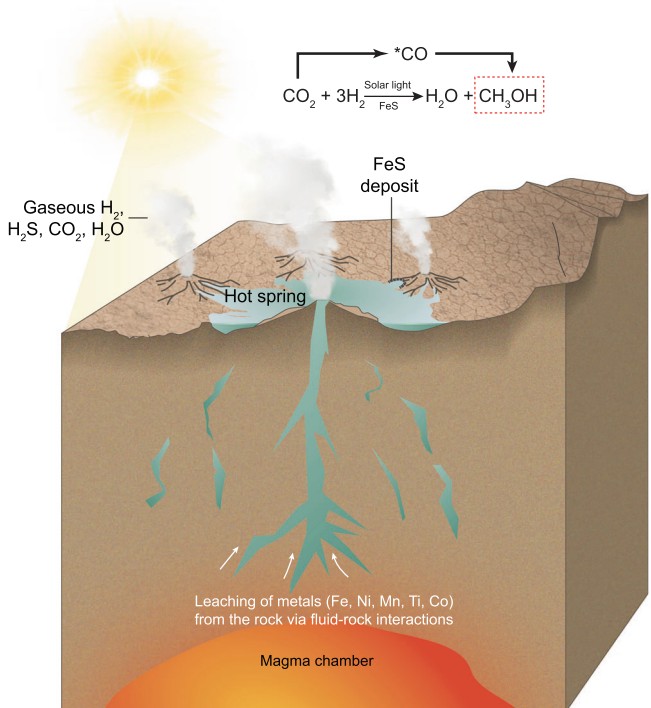

**Fig. 4 | Proposed terrestrial hot spring scenario.** FeS minerals deposited on the edges of hot spring pools catalyze the reduction of $CO_2$ driven by $H_2$ gas released from geochemical processes such as serpentinization. Light irradiation from the young Sun promotes the catalytic activity of FeS.

shown herein, irradiation (300–720 nm) was demonstrated to be optimal for the catalytic activities of the iron sulfides tested.

While $H_2$ remains essential in the proposed hot spring scenario, the present study reveals that FeS can convert $CO_2$ to $CH_3OH$ without the need for an aqueous phase, thereby eliminating the requirement for dissolved $H_2$ and a steep pH gradient suggested to be necessary to drive carbon fixation in deep-sea hydrothermal vents[12,13,15]. Using a continuous microfluidic setup, Hudson and coworkers[15] demonstrated the fixation of $CO_2$ to formate under moderate pressures (1.5 bar) and room temperature. During this process, dissolved $H_2$ in alkaline hydrothermal fluids donate electrons across the Ni-doped FeS membrane, reducing dissolved $CO_2$ in acidic prebiotic seawater. The lack of reduced carbon products in similar studies[12,13] was attributed to the loss of dissolved $H_2$ due to volatilization. Hudson and coworkers[15] also noted that a pH gradient of at least ~5 units is necessary to produce formate. Trapp and coworkers[55] demonstrated that a mixture of mackinawite (FeS) and dissolved $H_2S$ reduces KCN and other $C_1$ organic substrates into prebiotically relevant compounds, including thiols and methane, using batch conditions. Preiner and coworkers[14] showed that greigite ($Fe_3S_4$) suspended in water can promote organic synthesis when incubated under two bars of $H_2/CO_2$ (80/20) at 100 °C in a closed vessel, producing $C_1$ organic molecules (formate) at a rate of less than 0.1 $\mu mol\ g^{-1}\ min^{-1}$. For comparison, our study shows that pure FeS can catalyze the reaction of $H_2$ with $CO_2$ (15/5) under light in gaseous conditions, achieving a $C_1$ (i.e., methanol) production rate of 1.4 $\mu mol\ g^{-1}\ min^{-1}$ at 100 °C, which is ten times higher than that reported by Preiner and coworkers[14]. The ability to facilitate reduction reactions under ambient pressures without the need for aqueous solvation not only broadens the scope of potential prebiotic chemical environments for carbon fixation but also highlights the robust nature of FeS as catalysts under widely varying environmental conditions consistent with plausible early Earth scenarios. Together with these previous findings[14,55], our study suggests that ancient terrestrial hot springs may

have been able to support diverse FeS-catalyzed reduction mechanisms for various $C_1$ feedstocks in both aqueous and gaseous phases.

Understanding the parallels and disparities between the outcomes of this research and the acetyl-CoA pathway offers an additional critical perspective for assessing the hypothesis that the link between the acetyl-CoA pathway and iron sulfides is a direct consequence of prebiotic chemistry. Similar comparisons have been made in previous studies on submarine alkaline vents to support not only this idea but also the hypothesis that life originated at deep-sea hydrothermal vents[2-4,14-18,20-22]. In the acetyl-CoA pathway, one $CO_2$ molecule is converted to CO while another $CO_2$ molecule undergoes a six-electron reduction to form $CH_3$-$H_4$folate[7]. This methyl group is then transferred to the corrinoid iron-sulfur protein before combining it with CO and CoA to form acetyl-CoA[7]. Our study revealed FeS-catalyzed fixation of $CO_2$ to $CH_3OH$, the methyl group of which possesses the same oxidation state as $CH_3$-$H_4$folate. Mechanistic analysis suggests that this reduction employs a similar mechanism to that of the CODH/ACS enzyme[56], namely, a reverse water-gas shift with CO as a key intermediate. In contrast, $CO_2$ reduction herein was catalyzed by dry mackinawite under humid conditions, while the acetyl-CoA pathway enzymes function in the aqueous phase, with their [4Fe4S] clusters often linked to greigite[9]. Although such comparisons can offer intriguing suggestions about potential chemical evolutionary pathways, caution must be taken when extrapolating prebiotic chemistry to biology[57]. Nevertheless, the fact that $CO_2$ reduction mediated by FeS catalysis can take place in both submarine (aqueous) and terrestrial (gaseous) hydrothermal conditions would seem more in support of FeS clusters and the acetyl-CoA pathway being ancient—as opposed to the idea that life originated under one geological scenario or the other.

Although it is not exactly known how long-lasting or transient conditions like $H_2$, $CO_2$, heat, and light were in early hot spring environments, if these conditions did co-exist on early Earth, the reduction of $CO_2$ would likely have been driven forward. Whatever the case, our study contributes to the repertoire of potential prebiotic carbon fixation mechanisms in terrestrial hot spring environments. This conclusion is consistent with the emerging perspective that such settings, with their unique geochemical conditions and energy sources like sunlight, could have played a significant prebiotic chemical role during life's emergence.

## Methods

### FeS synthesis

Before starting the synthesis of FeS, all the apparatus involved were kept in an anaerobic chamber (Coy Laboratory Products) for a minimum of 48 h. The chamber was filled with a gas mixture of $N_2$/$H_2$/$CO_2$ (80/5/15) to remove any residual oxygen. Within this anaerobic environment, 0.3 M solutions of $Na_2S$ and $FeCl_2$ were prepared. FeS was simply synthesized by adding the $Na_2S$ solution to the $FeCl_2$ solution while stirring vigorously. The precipitated FeS solid was thoroughly washed with $N_2$-purged Milli-Q water for a minimum of six times and subsequently suspended in $N_2$-purged Milli-Q water to yield a 0.3 M FeS suspension. Prior to its use, $SiO_2$ powder (99.9% purity, Wako Pure Chem. Ind.) was pre-treated at 300 °C for 3 h under an air atmosphere to remove any surface-adsorbed organic contaminants. Inside the anaerobic chamber, the FeS suspension was mixed with the $SiO_2$ powder, and the mixture was stirred for 30 min. The mixed suspension was then dried at 25 °C for 48 h within the same anaerobic chamber. The dried solid was weighed and stored anaerobically in a glass test tube for further use. To prepare doped FeS, the dopant metals in salt form (i.e., $NiCl_2$, $MnCl_2$, $TiCl_2$, and $CoCl_2$) were dissolved in Milli-Q water to form 0.3 M solutions. These solutions were mixed with the 0.3 M $FeCl_2$ solution to achieve a 10% doping molar ratio. Then, the $Na_2S$ was added to form a solid precipitate, which was processed in the same manner as previously described for the preparation of FeS.

### Electron microscopy

Electron microscopic observations were conducted on the catalysts prior to the reaction to ascertain their crystalline structure. Specifically, the catalysts were retrieved from the anaerobic chamber using a gas transfer holder and subsequently introduced into the microscope load lock under vacuum conditions. The characterization and image acquisition of these catalysts were carried out using scanning/transmission electron microscopy (S/TEM) on a JEOL F200 instrument (UNSW) at an acceleration voltage of 200 kV.

### Photo-assisted thermocatalytic $CO_2$ hydrogenation

To examine the catalytic behavior of diverse metal-doped FeS catalysts under moderate temperatures (80–120 °C) and light irradiation, $CO_2$ hydrogenation experiments were conducted in a custom-built flow reactor at ambient pressure (Fig. S4). Normally, an LA-251 Xe lamp, equipped with a HA30 optical filter, was used to irradiate the catalysts with UV–visible light ($300 < \lambda < 720$ nm). For UV-enhanced experiments, a Hg-Xe LC8 L9566-01A lamp (Hamamatsu) (with emission of $200 < \lambda < 600$ nm) was used without any optical filter (Fig. S5). The light was directly transmitted onto the catalyst surface through a quartz window, with an irradiated area of 8.5 mm in diameter. A resistive heater, managed by a temperature control system, was used to balance the photothermal heating effect and maintain the desired catalyst temperatures. For each experiment, 10 mg of the catalysts was spread evenly on the reactor's reaction bed. Gaseous reactants of $CO_2$ and $H_2$ were then introduced at a 1:3 volume ratio into the reactor, at a total flow rate of 20 mL min$^{-1}$. After that, the reactor temperature was increased to the desired value. For experiments with water vapor, $CO_2$ and $H_2$ were introduced into the reaction chamber through a bubbler system containing ultrapure water.

A Shimadzu GC-2014 equipped with a TSG-1 15% SHINCARBON A 60-80 column (Shinwa Chemical Industries) was used for GC-FID analysis. 0.5 mL of the gaseous outflow from the reaction chamber was injected into the GC. The temperature of the column was kept at 80 °C from 0 to 10 min, then the temperature was raised to 120 °C at the rate of 40 °C min$^{-1}$ and held at 120 °C for 5 min. The carrier gas was argon with a flow rate of 25 mL min$^{-1}$.

### In situ diffuse reflectance infrared Fourier transform spectroscopy (DRIFTS) measurements

In order to elucidate the reaction intermediates and investigate the function of photo-induced charge carriers, an FT-IR-6300 system (JASCO Corp.) was utilized. This system features an in situ diffuse reflectance (DR) cell and a mercury–cadmium–telluride detector, cooled by liquid nitrogen. The DRIFTS measurements required the dispersion of 5 mg of catalysts within the DR cell, which was introduced to a 1:3 volume ratio of $CO_2$ and $H_2$. Once an adsorption-desorption equilibrium between the catalysts and reaction gas was achieved at a steady state at 120 °C, the system's background spectrum was recorded. The DR cell was then isolated by halting the reaction gas inflow, and DRIFTS spectra at specific time intervals to identify intermediates in the purely thermocatalytic process were recorded. Subsequently, light energy was introduced into the DR cell, and DRIFTS spectra were recorded at certain time intervals to study the role of photo-induced charge carriers in the reaction.

### Temperature-dependent kinetics of $CH_3OH$ production

To elucidate the temperature-dependent kinetics of $CH_3OH$ production, an Arrhenius plot was constructed based on the well-established Arrhenius equation:

$$\ln r = -\frac{E_a}{RT} + \ln A \qquad (1)$$

where $r$ denotes the rate of the reaction, $E_a$ is the activation energy, $R$ is the universal gas constant ($8.314 \, J \, mol^{-1} \, K^{-1}$), $T$ represents the absolute temperature, and $A$ is the pre-exponential factor. Reaction rates at various temperatures were first determined experimentally for $CH_3OH$ production. These rates were then natural-logarithm transformed and plotted against the inverse of the absolute temperature ($1/T$). The slope of the resulting linear plot yields ($-E_a/R$), which allows the direct determination of the activation energy $E_a$. Meanwhile, the intercept of this plot provides $\ln A$, from which the pre-exponential factor $A$ can be derived. By employing this method, the temperature dependence of the reaction kinetics was quantitatively characterized, offering insights into the underlying mechanistic processes of $CH_3OH$ production.

### Quantum mechanical modeling

First-principles calculations were carried out under the scheme of spin-polarized density functional theory (DFT) using the Vienna Ab-initio Simulation Package (VASP)[58,59]. FeS (100) surface was used to build the slab, and each slab consisted of six layers with $3 \times 2$ supercells. Two top-layer Fe atoms were replaced by Mn atoms in the FeS (100) slab to build the Mn-doped FeS (100) slab. In the geometry optimization, half of the slab atoms were fixed, and others were relaxed. The exchange-correlation interactions were described by generalized gradient approximation (GGA)[60] with the Perdew–Burke–Ernzerhof (PBE) functional[61]. Spin-polarization was included in all the calculations, and a damped van der Waals correction was incorporated using Grimme's DFT-D3 (BJ) scheme to better describe the non-bonding interactions[62]. The cut-off energies for plane waves were set to be 500 eV, and the residual force and energy on each atom during structure relaxation were converged to $0.005 \, eV \, Å^{-1}$ and $10^{-5} \, eV$, respectively. The Brillouin zones were sampled with a $k$-point mesh of $3 \times 3 \times 1$. The vacuum layer was -15 Å to remove the slab interaction between the z direction.

The transition states for the $*CO + H_2 \rightarrow *CHO + H$ step were calculated using the climbing-image nudge elastic (CI-NEB) method as implemented in the VASP transition state tools[63,64]. CI-NEB is an efficient method for determining the minimum energy diffusion path between two given positions. Twelve images, including initial and final positions, were employed for CI-NEB calculations. The atomic positions and energy of the images were relaxed until the largest norm of the force orthogonal to the path was smaller than $0.01 \, eV \, Å^{-1}$.

Gibbs free energies for each gaseous and adsorbed species were calculated at 300 K, according to the equation:

$$G = E_{DFT} + ZPE + \int C_v dT - TdS \qquad (2)$$

where $E_{DFT}$ is the electronic energy calculated with VASP, ZPE is the zero-point energy, $\int C_v dT$ is the enthalpy contribution and $TdS$ is the entropy contribution. Standard ideal gas methods were employed to compute $ZPE$ and $TdS$ from temperature, pressure, and the calculated vibrational energies. The details of $ZPE$, $\int C_v dT$, and $TdS$ are listed in Table S3 in Supplementary information.

### Assessment of experimental contamination

In the preparation phase of FeS catalysts, meticulous care was taken to exclude any organic materials from the process, followed by extensive washing to ensure the removal of potential contaminants. The $SiO_2$ powder (99.9% purity), which served as an inert support while promoting the dispersion of the FeS, was pre-treated at 300 °C for 3 h to remove any surface-adsorbed organic contaminants prior to its use. To verify the absence of organic contamination during experiments, a set of control blank experiments were conducted. Specifically, rigorous thermal treatments were applied at 120 °C under an argon (Ar) atmosphere, and pure $H_2$ without $CO_2$,

respectively (Supplementary Fig. S6), yielding no discernible by-products—evidence that supports the purity of the catalysts. Additionally, analytical investigations via DRIFTS have provided clear evidence of the dynamic changes in the reaction intermediates. These spectroscopic observations unambiguously indicate the production and subsequent depletion of intermediates, ultimately leading to the formation of $CH_3OH$, thus highlighting the catalytic behavior of the prepared FeS.

### Statistics

Unless specified otherwise, all experiments were conducted using three replicates ($N = 3$).

## Data availability

All data are available in the main text or the supplementary information. Source data are provided with this paper.

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

## Acknowledgements

We thank Kensuke Igarashi (National Institute of Advanced Industrial Science and Technology, Japan) for the valuable contributions, and Martina Preiner (Max Planck Institute for Terrestrial Microbiology) for providing critical comments. J.N. acknowledges support from the National Natural Science Foundation of China (NSFC; 42472303, 42106069), and the State Key Laboratory of Palaeobiology and Stratigraphy (223120). W.-N.L. acknowledges support from NSFC (42103016) and the State Key Laboratory of Nuclear Resources and Environment (2020Z20). A.C.F. acknowledges support from the University of New South Wales Strategic Hires and Retention Pathways (SHARP) program, the Australian Research Council Discovery Project Grant DP210102133, and a Future Fellowship, FT220100757. J.Y. acknowledges the support from the World Premier International Research Center Initiative (WPI) on Materials Nanoarchitectonics (MANA), MEXT (Japan), and the Photo-excitonix Project at Hokkaido University. The authors acknowledge the facilities and the scientific and technical assistance of Microscopy Australia at the Electron Microscope Unit (EMU) within the Mark Wainwright Analytical Centre (MWAC) at UNSW Sydney.

## Author contributions

J.N., S.L., and Q.P.T. conceived the research. S.L., Q.P.T., and W.-N.L. conducted experiments. Y.H. performed DFT calculations. J.N., S.L., Q.P.T., A.C.F., W.-N.L., Z.Y., J.Y., and M.J.V.K. performed research. J.N., S.L., and Q.P.T. wrote the manuscript with the input of all co-authors.

## Competing interests

The authors declare no competing interests.
