## [Peer Review File · Nature Communications]

Iron sulfide-catalyzed gaseous CO₂ reduction and prebiotic carbon fixation in terrestrial hot springs

Corresponding Author: Dr Shunqin Luo

Version 0:

Reviewer comments:

Reviewer #1

(Remarks to the Author)

Nan et al. reported some interesting results on reduction of CO₂ catalyzed by FeS mineral surface and its derivatives. The authors then conducted elaborate analyses of the minerals and designed theoretical simulations to decipher the mechanisms.

The results are solid with plenty of evidence. The writing reads well. I have some concerns before making final decisions:

1. Relevance with the primitive conditions:

(1) There are several experimental settings seemed to have low relevance with the early Earth. For example, the experiments used very high ratio of H₂/CO₂, but in primitive atmosphere, H₂ has a much lower mixing ratio than CO₂ (at least 3 orders of magnitude difference). Another example is the high M/Fe ratio (M = other transition metals used in the synthesis reactions). In fact, M should have much lower concentrations than Fe(II).

(2) Light: the authors focused on 300-720 nm light in most experiments. However, in the absence of ozone layer, UV light of 200-300 nm could also penetrate through the early atmosphere and reach the surface. You would need to have high levels of SO₂ in order to totally filter out most of the light < 300 nm, which is not relevant with the primitive Earth.

2. Importance of abiotic synthesis catalyzed by FeS at least in the way discussed here:

The authors conducted experiments mainly under dry conditions with some under water-vapor conditions. However, the land was more likely smaller with much larger coverage of ocean. Even if there were hot springs before (given the higher mantle temperature), most of it would be filled with liquid water. I would suggest the authors to conduct some further explorations in liquid fluid (with relevant compositions). A comparison of organic yield between this study and other potential sources is also helpful for the readers to understand the importance of this pathway.

Some line-by-line comments:

(1) Line 80: again, if you use more UV light (200-300 nm), would it have detrimental effects on organics as well as the mineral catalysts?

(2) Line 94: what is the pH of your solution? The real hot spring water contains high concentration of silica. It would be interesting to see its effect as well.

(3) Line 111-113: Is your measurement consistent with what people reported before?

(4) Line 164-165: again, this is probably more relevant with the primitive environment.

(5) Lines 198-201: given that CO is the key intermediate, if there is liquid water in your experiment, would simple organic acids, like HCOO⁻, become the predominant product? Like the previous study.

(6) Line 294-296: This comparison is weak as hot spring was likely much less abundant than hydrothermal vents on the

primitive Earth. Also, your dry experiment is even less than aqueous phase reactions.

(7) Line 505-507: This is not relevant with early water chemistry.

(8) Line 574-576: is the elemental ratio same to the value used in the experiment?

(9) Fig 2: Fe(Mn)S dark could yield organics as well. This is very interesting, but I'm curious about the effect of Fe:Mn ratio on this result.

Reviewer #2

(Remarks to the Author)

This experimental study extends origin of life theories relevant to alkaline vents into a hot spring setting, which links together aspects of prebiotic chemistry that are sometimes thought to be mutually exclusive; this is an important advance for origin of life studies. The complimentary DFT work will serve as an excellent reference. Described synthesis methods – largely based on hydrothermal materials and colloidal science synthesis techniques – are simple, facile, and complimentary to existing methods used in the field and related fields. A significant strength of the paper is that much detail is provided such that the study can be replicated. Even the custom built catalytic apparatus is well documented, photographed, and a schematic provided enabling reconstruction if desired. The potential significance of the work, at least in the case of Fe_{1-x}(Mn_x)S, is clearly articulated and well justified for its connections to prebiotic chemistry. The work mostly supports the conclusions and claims, but some additional supporting evidence is needed as described in the detailed report below. The methodology is largely sound; however some critical revisions are needed to fully meet expectations, particularly with respect to strong claims re: catalytic activity. At minimum, it is essential that particle size (average and distribution), band gap analysis, and bulk characterization is conducted. As well, an MnS control is a necessary comparison. While there do not appear to be fundamental flaws in the experiments conducted, incomplete analysis and missing controls limit the interpretation of results, integrity of the discussion, and limits further review capacity. The novelty claims are not quite a fair reflection of the state of the catalyst literature but perhaps do reflect earth & space, geochemical, and prebiotic chemistry literature pools; we recommend some additional discussion points and citations below. However, these do not necessarily prohibit publication and I believe the paper will be suitable after a major revision.

Detailed comments:

Claims of exclusive novelty need to be watered down to reflect the state of MnFeS alloys, solid solutions, and inter-metallics in catalysis literature, steel manufacturing / materials science, electrochemical and high temperature ceramics literature. One needn't go far to find well cited colloidal texts on applications, synthesis, and properties of MnS nanocrystals, let alone FeS systems with other transition metal dopants (e.g. Ferretti, Mondini, and Ponti, Chapter 6, *Advanced in Colloid Science*; and Chianelli, Daage, and Ledoux (1994). *Fundamental studies of transition-metal sulfide catalytic materials. Advances in catalysis*, 40, 177-232). This reviewer encourages the authors to review and acknowledge materials science, steel/ceramic processing, and electrochemical/catalysis journals for sulfidized Mn/Fe and other transition metal species. The novelty in this field and potential for new applications in prebiotic chemistry/origins of life study is clear and present, but the novelty of these conclusions is neither exclusive nor without context of catalytic and materials processing fields.

The proposed work is well thought out, thorough, and presents sound conclusions based on high quality data obtained with appropriate techniques, reasonable analysis methods, and appears replicable (though the data set provided, N=3, is small). The conclusions are defensible from the standpoint of the data presented though greater effort could be taken to dedicate analysis and explanation of potential mechanisms to the other dopant systems beyond Fe_{1-x}(Mn_x)S and the FeS control. Some work on this is presented and while the data included supports core conclusions and is sufficiently high quality to be worthy of publication, some of this data (i.e. for Fe_{1-x}(A_x)S for A = Ni, Co, Ti) requires greater extent of analysis, interpretation of possible mechanisms or, where this is beyond the scope or excessively cumbersome, at least a discussion on possible contributions to why lower catalytical activity was observed in these systems and their relative level of catalytic performance c.f. control FeS samples. Below we detail the laboratory analysis that we feel would be required for the claims in the paper to be sufficiently supported.

First, a control is required for a pure MnS species. This is an imperative test case to understand baseline behaviour of the two primary end-member species tested. Ideally this should occur for all other transition metals explored, including Co, Ni, and Ti, but at minimum for publication it is the opinion of this reviewer that a pure MnS control is necessary.

Further bulk characterization of the catalytic powders is necessary. TEM – while valuable for local nanoparticle crystallinity, solid-solution incorporation of Mn(II) into the Fe(II)S crystal structure, and the likes – is inherently local. A set of bulk powder characterization techniques is required:

- Bulk composition of nano-powders from PXRD is needed to determine whether there are independent crystallites of MnS or Mn_{1-x}Fe_xS together with Fe_{1-x}Mn_xS. Distributions of solid solution composition here will make a difference. If anoxic measures are challenging, I recommend encapsulation in Kapton pouches. Ideally this PXRD should be run before and after catalysis to validate crystal structures are unchanged. Rietveld refinement should be attempted to estimate mean particle sizes and variance from peak intensity and breadth as well as presence of minor phases and preferred surface textures.
- BET active surface area measurements are required in order to be able to compare the catalytic activity between FeS and Fe_{1-x}Mn_xS. BET values ideally will corroborate particle size estimates from PXRD, but will also provide distribution

information that could be pertinent. This is essential, because in catalysis (from Fischer-Tropsch to RWGS reactions) the preferred mechanisms, favouring of competitive pathways (e.g. carbon path vs hydrogen path in RWGS reactions), and concentration of active reaction intermediates all very strongly depend on reaction temperature, support composition (in this case silica), and active catalytic species' particle size. If the size distributions are similar between FeS and Fe_{1-x}Mn_xS then the comparison of catalytic activity is fair.

The discussion of band gap analysis needs refinement, comparison to known literature values, and improved comparison to the authors' own DFT results. How do your measurements of band gap compare to expected scaling by Vegard's law, in a compound of similar size and charge species and capable of solid solution formation, the compound's unit cell parameters shift linearly. This will be important because the band gap of FeS is smaller than MnS in pure materials; it is expected that the substitution of MnS into FeS would result in an increase in band gap (i.e. if Fe were added to MnS band gap would decrease). You are reporting the opposite with strong data, so why is this? Is the decrease in bandgap a result of carrier concentrations increasing or defect site increases that affect the band gap energy or direct vs. indirect nature of it?

• Some reference data and optical results are found in Tigwere, G. A., et al. (2023; *Materials Science in Semiconductor Processing*, 158, 107365) for Fe incorporation into MnS, but the reverse arguments can be made quite strongly too. Similar style of combined experimental-computational bandgap analysis is done by Qin et al. in the study of MnS/BaS catalysts for instance (*ACS Appl. Mater. Interfaces* 2024, 16, 16, 20463–20471).

In the discussion: when referencing the Hudson et al. 2020 PNAS study, it should be referred to as Hudson and coworkers (as it was Hudson's lab that led the study).

The citations are mostly complete but several recent prebiotic chemistry references about iron-sulfide driven CO₂ reduction are missing; these should also be mentioned as to how this work compares to their results:

- Santos-Carballal David, Roldan Alberto, Dzade Nelson Y. and de Leeuw Nora H. 2018; Reactivity of CO₂ on the surfaces of magnetite (Fe₃O₄), greigite (Fe₃S₄) and mackinawite (FeS) *Phil. Trans. R. Soc. A* 376:20170065
- Roldan et al., Bio-inspired CO₂ conversion by iron sulfide catalysts under sustainable conditions. *Chem. Commun.*, 2015, 51, 7501-7504
- Yamaguchi et al. 2014, Electrochemical CO₂ Reduction by Ni-containing Iron Sulfides: How Is CO₂ Electrochemically Reduced at Bisulfide-Bearing Deep-sea Hydrothermal Precipitates? *Electrochimica Acta* 141 311-318

Reviewer #3

(Remarks to the Author)

Reviewer #4

(Remarks to the Author)

Version 1:

Reviewer comments:

Reviewer #1

(Remarks to the Author)

I am re-reviewing this paper and would like to thank the authors for their responses to my initial comments. However, I do not think the authors addressed my primary concerns raised in my prior review.

The largest concern of my previous review is the relevance and importance of these processes on the primitive Earth. I agree with the authors that the hot spring system was critical for the origin of life in terms of wet-dry cycling etc. However, it is also an open system with quick diffusion of gases into the early atmosphere. Otherwise, you cannot have enough CO₂ to sustain the reaction. The early atmosphere contained much lower H₂ (<10⁻³ atm) than the values you used in the experiment. Thanks for the supplementary experiments testing the effect of H₂/CO₂. If we extend the trend to lower H₂/CO₂ like the primitive atmosphere, the yield would be much lower. I also agree with the authors that some enclosed bubbles in hot spring might be H₂-rich, although please make sure that the cited hydrothermal experiments are not relevant here. But, your experiment was conducted with a much higher dose (per m² per second) of UV light than the natural settings (with frequent day/night cycle) and it still required minutes to have some noticeable yield. In nature, you cannot find such a perfect environment relevant to your experimental settings, where high dose of UV (requires shallow and thus open), enough H₂ (requires deep and enclosed), and enough reaction time (requires deep and enclosed as well) were all present because

they are in conflict per se.

In the revised manuscript, I was also expecting to see a at least semi-quantitative comparison between the proposed process and other mainstream hypotheses, such as the cometary delivery, photochemistry in the air, and deep-sea hydrothermal systems. It seems now that this proposed pathway is perhaps trivial compared with other processes, given the concerns raised above.

Overall, I think the importance and relevance of this work are overstated. In my view, this manuscript is much better suited to a discipline-specific journal rather than a multidisciplinary journal like Nature Communications. I wish you success in your efforts to find a more appropriate venue for your work.

Reviewer #2

(Remarks to the Author)

I co-reviewed this manuscript with an early career researcher who provided the listed reports. This is part of the Nature Communications initiative to facilitate training in peer review and to provide appropriate recognition for Early Career Researchers who co-review manuscripts.

The authors have undertaken significant effort to address each point of concern from this reviewer. The inclusion of BET data, bulk XRD compositional analysis for grain size estimates from the Scherer equation, and an MnS control make the conclusions all the more impactful while removing ambiguity in interpretation. Owing to these improvements to the text, I agree with the conclusions of the authors and supports the publication of this work. I thank the authors for addressing each review point in detail. While it is a shame that TEM could not be conducted, I believe the remaining edits and additional data reported are sufficient. The work is novel, the analysis sound, and is significant in extending alkaline vent theories of origins of life to those of hot spring settings. I recommend publication with no further revisions.

Reviewer #3

(Remarks to the Author)

Reviewer #4

(Remarks to the Author)

The point-by-point replies to the reviewers' comments

We sincerely appreciate the valuable comments and suggestions from the reviewers that help us to significantly improve our manuscript. For clearness reason, the responses were marked with **BLUE** color and the revision parts in the revised manuscript were marked with **RED** color.

Reviewer #1 (Remarks to the Author):

Nan et al. reported some interesting results on reduction of CO₂ catalyzed by FeS mineral surface and its derivatives. The authors then conducted elaborate analyses of the minerals and designed theoretical simulations to decipher the mechanisms. The results are solid with plenty of evidence. The writing reads well. I have some concerns before making final decisions:

Response: We thank the reviewer for the insightful review and positive feedback, particularly the insights concerning relevance of the experimental set up to early Earth conditions. We have now performed a series of rigorous supplemental experiments and added more discussion to address the reviewer's comments and concerns. Detailed responses are given below.

1. Relevance with the primitive conditions:

(1) There are several experimental settings seemed to have low relevance with the early Earth. For example, the experiments used very high ratio of H₂/CO₂, but in primitive atmosphere, H₂ has a much lower mixing ratio than CO₂ (at least 3 orders of magnitude difference). Another example is the high M/Fe ratio (M = other transition metals used in the synthesis reactions). In fact, M should have much lower concentrations than Fe(II).

Response: Thank you for raising this point. While the global primitive atmosphere may not have had a high H₂/CO₂ ratio, this work focuses on terrestrial hot springs where serpentinization likely gave rise to high H₂/CO₂ ratios. Specifically, Zgonnik et al. (2020) highlighted the natural occurrences of hydrogen and showed that hot springs undergoing serpentinization could contain 15–80 vol.% H₂, in contrast to other gases, including CO₂. Furthermore, previous research on submarine hydrothermal vents—environments analogous to hot springs affected by serpentinization—have also performed catalytic experiments using an H₂/CO₂ ratio of 80/20 (Preiner et al., 2020). Therefore, we consider that the H₂/CO₂ ratios of 15/5 used in our study is reasonable and reflective of these natural settings. Nevertheless, to strengthen the relevance of our experiments to primitive conditions, we have performed new experiments employing lower H₂/CO₂ ratios of 5/15 and 10/10. The results indicate that an increase in the H₂/CO₂ ratio leads to higher production of methanol (Supplementary Figure S8a). In the lowest ratio case (5/15), significant production of CH₃OH (up to 2.6 μmol g⁻¹ min⁻¹) can still be observed. These results suggest that CO₂ fixation could have happened under the varying H₂/CO₂ ratios that one might expect for an ancient terrestrial hot spring.

To test the influence of different Mn doping amounts, we have included new experiments where FeS was doped with Mn at 0.1%, 1%, and 10%, and pure MnS as a control (Supplementary Figure S8b). The production rate increases with higher Mn abundance, whereas pure MnS shows the lowest performance under both dark and light conditions. For all catalysts, UV–Vis irradiation increased methanol production. The results indicate the catalytic activity of FeS was significantly improved even at lower Mn-doping percentages. These experiments suggest that the methanol production mechanism discussed could take place in a diversity of hot springs, each containing different concentrations of Mn.

Action: Please see Line 169 in the revised main text: “To evaluate the robustness of CH₃OH production under different H₂/CO₂ ratios, we also conducted experiments at lower H₂/CO₂ ratios of 5/15 and 10/10 on Mn-doped FeS. The results show that an increase in the H₂/CO₂ ratio leads to higher production of CH₃OH (Supplementary Figure S8a). Even at the lowest ratio (5/15), however, considerable production of CH₃OH (up to 2.6 μmol g⁻¹ min⁻¹) can be observed.”

Please see Line 186 in the revised main text: “Further experiments employing various Mn percentages, specifically 0.1%, 1%, and 10%, show observable improvement of FeS catalytic activity even at lower Mn-doping percentages (Supplementary Figure S8b). The results indicate that in both dark and UV–visible conditions, CH₃OH yield increases with Mn percentages.”

Supplementary Figure S8a–b

(2) Light: the authors focused on 300-720 nm light in most experiments. However, in the absence of ozone layer, UV light of 200-300 nm could also penetrate through the early atmosphere and reach the surface. You would need to have high levels of SO₂ in order to totally filter out most of the light < 300 nm, which is not relevant with the primitive Earth.

Response: Thank you for raising this point. We have now added new experiments that include the 200–300 nm range, building on our previous research results. These experiments employed a Hamamatsu LC8 L9566-01A Hg-Xe lamp, which generates 200–600 nm light, in conjunction with a LA-251 Xe lamp equipped with an HA30 optical filter, which generates 300–720 nm light. Together, these two light sources closely resemble the average spectrum, which we have labelled as “UV-enhanced”, that likely reached the surface of early Earth in the absence of SO₂ and ozone (Gaier et al., 2010). The results are consistent with previous findings that enhancing lower wavelengths decrease the yield of methanol, see the figure below (please also see Figures 2d and S5).

Notwithstanding models of the early atmosphere indicating low global concentrations of SO₂ and H₂S due to photolysis and reactions with oxidants, these species may have exhibited higher local concentrations in environments closely associated with volcanic activity like the terrestrial hot springs proposed. As such, we consider it is plausible that ancient hot springs were shielded from lower wavelengths at least periodically if not for extended periods of time.

Action. We have included new experimental results in Supplementary Figure S8c and a brief discussion regarding SO₂ and H₂S concentrations in terrestrial hot springs and their potential shielding of lower UV wavelengths at **Line 333**: *“local concentrations of SO₂ and H₂S surrounding terrestrial hot springs may have been high enough to block lower wavelengths (200–300 nm), especially during periods of high volcanic activity.”*

Supplementary Figure S8c

2. Importance of abiotic synthesis catalyzed by FeS at least in the way discussed here:

The authors conducted experiments mainly under dry conditions with some under water-vapor conditions. However, the land was more likely smaller with much larger coverage of ocean. Even if there were hot springs before (given the higher mantle temperature), most of it would be filled with liquid water. I would suggest the authors to conduct some further explorations in liquid fluid (with relevant compositions). A comparison of organic yield between this study and other potential sources is also helpful for the readers to understand the importance of this pathway.

Response: Thank you for your suggestion. We agree that further exploring the aqueous phase chemistry of the FeS catalysts would allow for a more holistic assessment of the importance of CO₂ reduction pathway. Our paper focuses, however, on the gas-solid phase chemistry hypothesised to have occurred on the dry land segments of hot springs to fill a more immediate gap in the early Earth geochemistry literature. Although we did not conduct aqueous phase experiments, these types of studies have already been carried out extensively, and are discussed on Lines 293–308 in the initial manuscript. We have expanded this discussion (Lines 348 in the revised manuscript) to compare our data with relevant results from these papers. We want to note that Sojo et al. and Vasiliadou et al. have investigated CO₂ reduction driven by H₂ and a pH gradient under atmospheric pressures with both studies yielding negative results. The lack of reduced carbon products was attributed to the volatilization of H₂ under atmospheric pressures. Since our setup also employs a low-pressure open system where H₂ is unlikely to dissolve into an aqueous phase, we feel it is unlikely that H₂-driven CO₂ reduction would have occurred in solution under the terrestrial conditions we are modelling. Nevertheless, if enough H₂ is dissolved to facilitate H₂-driven CO₂ reduction, we hypothesize that the products will be similar to those previously shown

for greigite by Preiner and coworkers (2020). Moreover, although terrestrial land was likely limited in size, its advantages such as a mechanism for concentrating organics, i.e. wet-dry cycling, and access to energy sources (e.g. sunlight and lightning) and feedstocks (e.g. from photochemistry and interstellar dust) make terrestrial hot springs highly advantageous scenarios for life's origins.

Action: In the revised manuscript, we have revised the discussion from **Line 348–362** according to our above response: “*Trapp and coworkers demonstrated that a mixture of mackinawite (FeS) and dissolved H₂S reduces KCN and other C₁ organic substrates into prebiotically relevant compounds, including thiols and methane, using batch conditions. Preiner and coworkers showed that greigite (Fe₃S₄) suspended in water can promote organic synthesis when incubated under two bars of H₂/CO₂ (80/20) at 100 °C in a closed vessel, producing C₁ organic molecules (formate) at a rate of less than 0.1 μmol g⁻¹ min⁻¹. For comparison, our study shows that pure FeS can catalyze the fixation of H₂ with CO₂ (15/5) under light in gaseous conditions, achieving a C₁ (i.e., methanol) production rate of 1.4 μmol g⁻¹ min⁻¹ at 100 °C, which is ten times higher than that reported by Preiner et al. (2020). The ability to facilitate reduction reactions under ambient pressures without the need for aqueous solvation not only broadens the scope of potential prebiotic chemical environments for carbon fixation but also highlights the robust nature of FeS as catalysts under widely varying environmental conditions consistent with plausible early Earth scenarios. Together with these previous findings, our study suggests that ancient terrestrial hot springs may have been able to support diverse FeS-catalyzed reduction mechanisms for various C₁ feedstocks in both aqueous and gaseous phases.*”

Some line-by-line comments:

(1) Line 80: again, if you use more UV light (200-300 nm), would it have detrimental effects on organics as well as the mineral catalysts?

We have shown that enhancing the 200–300 nm region decreases the methanol yield, which is still, nevertheless, higher than that of the dark reactions at 100–120 °C. We have discussed this potential impact in detail in **Line 206**, saying that “*UV-enhanced irradiation was observed to have a detrimental effect on CH₃OH production. Methanol production upon UV-enhanced irradiation was below the detection limit at 80–90 °C but was observed for 100–120 °C (Figure 2d). This decrease in methanol yield may be a consequence of the higher energy UV light aiding in the decomposition of adsorbed CH₃OH or other intermediates formed on the surface of the catalysts before completion of the overall reaction cycle.*” The observed increase in catalytic behavior of Fe(Mn)S under UV-enhanced light irradiation compared to dark condition at 120 °C suggest that the catalyst is not affected significantly by the UV-enhanced light.

(2) Line 94: what is the pH of your solution? The real hot spring water contains high concentration of silica. It would be interesting to see its effect as well.

Thank you for raising this point. The pH of the FeS suspension is almost neutral (pH=7–8), as a consequence of the acidity of FeCl₂ and alkalinity of Na₂S. We indeed considered that real hot spring water contains a high concentration of silica, which is why we used pre-treated SiO₂ powder as an inert support to promote the dispersion of the FeS samples. However, since the current paper focuses on the gas-solid interface, the effect of aqueous silica on CO₂ reduction falls beyond the scope of our investigation.

(3) Line 111-113: Is your measurement consistent with what people reported before?

Yes, our results (a band gap of ~2.5 eV for FeS) are consistent with previous measurements, which reported that the synthesized FeS nanoparticles analyzed by optical absorbance exhibited a direct energy band gap of 2.39 eV. Moreover, we observed a slight decrease in the band gap upon the introduction of Mn into FeS. This aligns well with previous reported trends (Yaqoob et al., 2022). In their study, the narrowing of the band gap can be attributed to the interaction between electrons and the introduced impurity levels, as well as the increase in charge carrier concentration caused by Mn doping.

(4) Line 164-165: again, this is probably more relevant with the primitive environment.

Thank you for this comment. We agree that the hot spring vapor-zone conditions in the primitive environment would have included water vapor and potentially greater exposure to UV light in the 200–300 nm range. We have now included experiments that include this wavelength region. Please see our response above for more details.

(5) Lines 198-201: given that CO is the key intermediate, if there is liquid water in your experiment, would simple organic acids, like HCOO⁻, become the predominant product? Like the previous study.

Thank you for your question. The presence of liquid water could indeed influence the reaction pathway and the resulting products. As discussed above, it is unclear how much H₂ can dissolve into the aqueous phase. Assuming that enough H₂ is dissolved to facilitate CO₂ reduction, the presence of solvent molecules like water is expected to significantly alter the ensuing chemistry, e.g., its reaction with CO to yield formate. In a liquid medium, the O atom of CO₂ may interact more strongly with the FeS surface, as the O atom carries a partial negative charge and can form stronger interactions with positively charged sites on the surface. This type of O atom adsorption

might be more favorable for the production of formate compared to conditions described in this work where CO might instead be converted to other products like methanol or longer-chain hydrocarbons.

(6) Line 294-296: This comparison is weak as hot spring was likely much less abundant than hydrothermal vents on the primitive Earth. Also, your dry experiment is even less than aqueous phase reactions.

Although terrestrial land was likely limited in size, its advantages such as a mechanism for concentrating organics, i.e. wet-dry cycling, and access to energy sources (e.g. sunlight and lightning) and feedstocks (e.g. from photochemistry and interstellar dust) make terrestrial hot springs highly advantageous scenarios for life's origins. In addition, we have highlighted (Lines 348–356) that our gas phase experiment shows FeS (mackinawite) reducing CO₂ more efficiently than Fe₃S₄ (greigite) in the aqueous phase experiments conducted by Preiner and coworkers under otherwise comparable conditions.

(7) Line 505-507: This is not relevant with early water chemistry.

We recognize that the specific ratios of metals other than iron, as used in our experiments, may not necessarily reflect all geological environments on ancient Earth. However, the primary goal of this study was to supplement the prebiotic chemistry literature by exploring the potential of FeS catalysts and compare their activities for abiotic carbon fixation under dry conditions. To accommodate a wider range of Mn availability, we have conducted more experiments comparing pure FeS, pure MnS, and Mn-doped FeS at 0.1%, 1%, and 10%. For more discussion, please see our previous response.

(8) Line 574-576: is the elemental ratio same to the value used in the experiment?

During the DFT calculations, we used a slab consisting of six layers with a 3 x 2 supercell to model the role of Mn-doping in CO₂ hydrogenation, corresponding to approximately 10% Mn doped in FeS, i.e., the same as we used in the experiments.

(9) Fig 2: Fe(Mn)S dark could yield organics as well. This is very interesting, but I'm curious about the effect of Fe:Mn ratio on this result.

We have included Mn-doped FeS at 0.1%, 1%, and 10% to show the effect of Fe:Mn ratios. The results show that the production rate of methanol increases with the percentage of Mn doping in FeS. Under both dark and UV-Vis light conditions, the methanol production rate peaks with 10% Mn

doping in FeS. The results highlight the significant role of Mn in enhancing the catalytic activity of FeS for CO₂ to CH₃OH conversion.

Action: Please see **Line 186** in the revised manuscript: “*Further experiments employing various Mn percentages, specifically 0.1%, 1%, and 10%, show observable improvement of FeS catalytic activity even at lower Mn-doping percentages (Supplementary Figure S8b). The results indicate that in both dark and UV-Vis conditions, CH₃OH yield increases with Mn percentages.*”

Reviewer #2 (Remarks to the Author):

1. This experimental study extends origin of life theories relevant to alkaline vents into a hot spring setting, which links together aspects of prebiotic chemistry that are sometimes thought to be mutually exclusive; this is an important advance for origin of life studies. The complimentary DFT work will serve as an excellent reference. Described synthesis methods – largely based on hydrothermal materials and colloidal science synthesis techniques – are simple, facile, and complimentary to existing methods used in the field and related fields. A significant strength of the paper is that much detail is provided such that the study can be replicated. Even the custom built catalytic apparatus is well documented, photographed, and a schematic provided enabling reconstruction if desired. The potential significance of the work, at least in the case of $\text{Fe}_{1-x}(\text{Mn}_x)\text{S}$, is clearly articulated and well justified for its connections to prebiotic chemistry. The work mostly supports the conclusions and claims, but some additional supporting evidence is needed as described in the detailed report below. The methodology is largely sound; however some critical revisions are needed to fully meet expectations, particularly with respect to strong claims are: catalytic activity. At minimum, it is essential that particle size (average and distribution), band gap analysis, and bulk characterization is conducted. As well, an MnS control is a necessary comparison. While there do not appear to be fundamental flaws in the experiments conducted, incomplete analysis and missing controls limit the interpretation of results, integrity of the discussion, and limits further review capacity. The novelty claims are not quite a fair reflection of the state of the catalyst literature but perhaps do reflect earth & space, geochemical, and prebiotic chemistry literature pools; we recommend some additional discussion points and citations below. However, these do not necessarily prohibit publication and I believe the paper will be suitable after a major revision.

Response: We thank the reviewer for the supportive and comprehensive feedback on our study, and for recognition of the potential our research holds for advancing origin of life theories by integrating hot spring settings with alkaline vent chemistry. We have addressed all issues in the revised manuscript by including more detailed particle size analysis (Supplementary Table S1), and bulk characterization by XRD (Supplementary Figure S3). Specifically, we find that the FeS and Mn-doped FeS catalysts show very similar specific surface areas ($167.8 \text{ m}^2 \text{ g}^{-1}$ and $166.4 \text{ m}^2 \text{ g}^{-1}$, respectively), indicating that the Mn indeed serves as a chemically reactive site for promoting catalytic behavior. We further utilized the Scherrer equation to evaluate the mean size of FeS and Mn-doped FeS nanocrystals, finding that the introduction of Mn into FeS crystals slightly increases crystallinity (FeS: 12.2 nm; Fe(Mn)S: 13.0 nm). It should be noted that Mn-doped FeS formed as irregular or plate-like aggregates (Figure 1a), composing of nanometer-sized small crystals. The

size of these plate-like nanocrystals ranged from several to tens of nanometers, however, distinguishing individual crystals within the aggregates from TEM observation remains challenging.

We also have included a pure MnS control in the revised manuscript, and extend this approach to include Mn-doped FeS at 0.1%, 1%, and 10%. The results show that the production rate of methanol increases with the percentage of Mn doping in FeS. Under both dark and UV-Vis light conditions, the methanol production rate is highest with 10% Mn doping in FeS. In contrast, pure MnS exhibits the lowest production under both dark and light conditions, although it still demonstrates some enhancement due to light irradiation. Previous studies indicated that MnS has a restricted capacity for CO₂ adsorption and activation, which hinders its effectiveness as a catalyst for CO₂ reduction (Li et al., 2021). It is therefore likely that the insufficient adsorption sites and weak interaction with CO₂ molecules reduce the catalytic potential, making MnS less efficient compared to FeS or Mn-doped FeS. Furthermore, MnS typically exhibits a wide band gap (~3.2 eV), making it primarily responsive to UV light (Ferretti et al., 2016). However, exposure to enhanced-UV light can be detrimental to the as-formed CH₃OH products, potentially leading to their degradation. In contrast, FeS and Mn-doped FeS are UV-visible light responsive, allowing for more efficient CO₂-to-CH₃OH conversion. Consequently, MnS shows only a slight enhancement in CO₂-to-CH₃OH conversion under UV-visible light conditions.

Action: In the revised manuscript **Line 186**: “Further experiments employing various Mn percentages, specifically 0.1%, 1%, and 10%, show observable improvement of FeS catalytic activity even at lower Mn-doping percentages (Supplementary Figure S8b). The results indicate that in both dark and UV-Vis conditions, CH₃OH yield increases with Mn percentages. In contrast, experiments with pure MnS showed the lowest activity under both dark and light conditions, with only slight enhancement from light irradiation. MnS has limited capacity for CO₂ adsorption and activation, hindering its catalytic effectiveness (Li et al., 2021). Its wide band gap (~3.2 eV) makes it primarily responsive to UV light (Ferretti et al., 2016), which can degrade the as-formed CH₃OH products. Conversely, pure and Mn-doped FeS respond to both UV and visible light, resulting in more efficient CO₂-to-CH₃OH conversion.”

Supplementary Figure S8b

Detailed comments:

2. Claims of exclusive novelty need to be watered down to reflect the state of MnFeS alloys, solid solutions, and inter-metallics in catalysis literature, steel manufacturing / materials science, electrochemical and high temperature ceramics literature. One needn't go far to find well cited colloidal texts on applications, synthesis, and properties of MnS nanocrystals, let alone FeS systems with other transition metal dopants (e.g. Ferretti, Mondini, and Ponti, Chapter 6, *Advanced in Colloid Science*; and Chianelli, Daage, and Ledoux (1994). *Fundamental studies of transition-metal sulfide catalytic materials. Advances in catalysis*, 40, 177-232). This reviewer encourages the authors to review and acknowledge materials science, steel/ceramic processing, and electrochemical/catalysis journals for sulfidized Mn/Fe and other transition metal species. The novelty in this field and potential for new applications in prebiotic chemistry/origins of life study is clear and present, but the novelty of these conclusions is neither exclusive nor without context of catalytic and materials processing fields.

Response: Thank you for your insightful and detailed feedback on our manuscript. We greatly appreciate your suggestions for broadening the context of our study within the existing literature. In the revised manuscript, we have included the literature citation, including the fundamental studies of transition-metal sulfide catalytic materials and colloidal applications to ensure our discussion accurately represents the state of the field and acknowledges prior research.

3. The proposed work is well thought out, thorough, and presents sound conclusions based on high quality data obtained with appropriate techniques, reasonable analysis methods, and appears replicable (though the data set provided, $N=3$, is small). The conclusions are defensible from the standpoint of the data presented though greater effort could be taken to dedicate analysis and explanation of potential mechanisms to the other dopant systems beyond $\text{Fe}_{1-x}(\text{Mn}_x)\text{S}$ and the FeS control. Some work on this is presented and while the data included supports core conclusions and is sufficiently high quality to be worthy of publication, some of this data (i.e. for $\text{Fe}_{1-x}(\text{A}_x)\text{S}$ for $\text{A} = \text{Ni}, \text{Co}, \text{Ti}$) requires greater extent of analysis, interpretation of possible mechanisms or, where this is beyond the scope or excessively cumbersome, at least a discussion on possible contributions to why lower catalytical activity was observed in these systems and their relative level of catalytic performance c.f. control FeS samples.

Response: Thank you for your feedback and suggestion. We observed lower catalytic performance upon the introduction of Ni, Co and Ti. Theoretically, production of CH_3OH as final product of CO_2 hydrogenation requires optimized adsorption/desorption of reactants, critical intermediates, and products. In our experiments, we observed the production of CH_4 as one of the products using Ni-doped FeS, indicating that methoxy intermediates might be further hydrogenated to CH_4 before desorption (Sun et al., 2017), lowering the production of CH_3OH . Co-based materials are generally considered as efficient catalyst for Fischer-Tropsch process, considering its unique capacity for adsorption and partial dissociation of carbon species. Therefore, introducing Co into FeS system may retard the desorption of CH_3OH due to the strong interaction between Co active sites and CH_3OH (Lu et al., 2024). Similar phenomena may also exist in Ti-doping case, since the Ti sites could facilitate spontaneous dissociation of CH_3OH products through the formation of Ti-OCH_3 (Luo et al., 2023). All these unique characteristics lead to lower CH_3OH production and release for Ni-, Co-, and Ti-doped FeS compared to pristine FeS.

Action: In the revised manuscript, we have included the above discussion in **Line 148**: “Lower catalytic performance upon the introduction of Ni, Co, and Ti into the FeS system was observed. This result could be attributed to the fact that the production of CH_3OH as the final product of CO_2

hydrogenation requires optimal adsorption and desorption of reactants, critical intermediates, and products. In these experiments, the production of CH₄ was observed as one of the products when using Ni-doped FeS (Supplementary Table S2), suggesting that methoxy intermediates might be further hydrogenated to CH₄ before desorption, thereby reducing the yield of CH₃OH. Co-based materials are generally recognized as efficient catalysts for the Fischer-Tropsch process due to their strong capacity for the adsorption and partial dissociation of carbon species. Therefore, introducing Co into the FeS system may hinder the desorption of CH₃OH due to the strong interaction between Co active sites and CH₃OH. A similar phenomenon may also occur in the case of Ti-doping, where Ti sites could facilitate the spontaneous dissociation of CH₃OH through the formation of Ti-OCH₃, leading to lower CH₃OH production. These unique characteristics contribute to lower CH₃OH production and release for Ni-, Co-, and Ti-doped FeS compared to pure FeS.”

4. Below we detail the laboratory analysis that we feel would be required for the claims in the paper to be sufficiently supported. First, a control is required for a pure MnS species. This is an imperative test case to understand baseline behaviour of the two primary end-member species tested. Ideally this should occur for all other transition metals explored, including Co, Ni, and Ti, but at minimum for publication it is the opinion of this reviewer that a pure MnS control is necessary.

Response: Thank you for your constructive feedback. We agree that incorporating a control for a pure MnS species is essential. We have included a pure MnS control in our experiments and extend this approach to include Mn-doped FeS at 0.1%, 1%, and 10%. The results show that the production rate of methanol increases with the percentage of Mn doping in FeS. Under both dark and UV-Vis light conditions, the methanol production rate is highest with 10% Mn doping in FeS. These results highlight the significance of Mn doping in enhancing the photocatalytic reduction of CO₂ to methanol using FeS-based catalysts. However, pure MnS exhibits the lowest production rates under both dark and light conditions, although it still demonstrates some enhancement due to light irradiation. Previous studies indicated that MnS has a restricted capacity for CO₂ adsorption and activation, which hinders its effectiveness as a catalyst for CO₂ reduction (Li et al., 2021). It is therefore likely that the insufficient adsorption sites and weak interaction with CO₂ molecules reduce the catalytic potential, making MnS less efficient compared to FeS or Mn-doped FeS. Furthermore, MnS typically exhibits a wide band gap (~3.2 eV), making it primarily responsive to UV light (Ferretti et al., 2016). However, exposure to UV light can be detrimental to the as-formed CH₃OH products, potentially leading to their degradation. In contrast, FeS and Mn-doped FeS are responsive to both UV and visible light, allowing for more efficient CO₂-to-CH₃OH conversion.

Consequently, MnS shows only a slight enhancement in CO₂-to-CH₃OH conversion under UV-visible light conditions.

Action: Please see **Line 186** in the revised manuscript: *“Further experiments employing various Mn percentages, specifically 0.1%, 1%, and 10%, show observable improvement of FeS catalytic activity even at lower Mn-doping percentages (Supplementary Figure S8b). The results indicate that in both dark and UV-Vis conditions, CH₃OH yield increases with higher Mn percentages. In contrast, experiments with pure MnS showed the lowest activity under both dark and light conditions, with only slight enhancement from light irradiation. MnS has limited capacity for CO₂ adsorption and activation, hindering its catalytic effectiveness. Its wide band gap (~3.2 eV) makes it primarily responsive to UV light, which can degrade the as-formed CH₃OH products. Conversely, pure and Mn-doped FeS respond to both UV and visible light, resulting in more efficient CO₂-to-CH₃OH conversion”.*

5. Further bulk characterization of the catalytic powders is necessary. TEM – while valuable for local nanoparticle crystallinity, solid-solution incorporation of Mn(II) into the Fe(II)S crystal structure, and the likes – is inherently local. A set of bulk powder characterization techniques is required: Bulk composition of nano-powders from PXRD is needed to determine whether there are independent crystallites of MnS or Mn_{1-x}Fe_xS together with Fe_{1-x}Mn_xS. Distributions of solid solution composition here will make a difference. If anoxic measures are challenging, I recommend encapsulation in Kapton pouches. Ideally this PXRD should be run before and after catalysis to validate crystal structures are unchanged. Rietveld refinement should be attempted to estimate mean particle sizes and variance from peak intensity and breadth as well as presence of minor phases and preferred surface textures.

Response: Thank you for your valuable feedback on the need for comprehensive bulk characterization of the catalytic powders. We conducted XRD for bulk powder (without the SiO₂ support) using an airtight specimen holder to analyze the structures of the catalysts. The results suggest that FeS featured a short-range mackinawite ordering, manifesting an average crystalline size of 12.2 nm according to the Scherrer equation. Introducing Mn into FeS slightly increased the crystallinity (mean size: 13.0 nm), with the appearance of MnS in the solid solution. It should be noted that the observed catalytic enhancement in Mn-doped FeS samples should be mainly attributed to the formation of the Fe(Mn)S solid solution, considering the low activity and nearly negligible photo-enhancement of pure MnS catalysts.

Action: In the revised manuscript **Line 119**, “The X-ray diffraction (XRD) pattern of FeS featured a short-range mackinawite ordering, manifesting an average crystalline size of 12.2 nm according to Scherrer equation. Introducing Mn into FeS slightly increased the crystallinity, with the appearance of MnS in the Mn-doped FeS solid solution mixture (Supplementary Figure S3 and Table S1).

Response: Furthermore, we are aware that conducting XRD analyses both before and after the catalysis experiments helps to confirm whether the crystal structures remain unchanged throughout the catalytic processes. Since we used SiO₂ powder as an inert support to promote the dispersion of the Fe(Mn)S samples during activity measurement, the XRD analysis of spent Fe(Mn)S shows signals for amorphous SiO₂ without obvious diffraction peaks for crystallized Fe(Mn)S, suggesting that there is no significant recrystallization or grain growth of Fe(Mn)S after catalysis (Figure R1).

Figure R1. XRD pattern for spent Fe(Mn)S catalyst.

6. BET active surface area measurements are required in order to be able to compare the catalytic activity between FeS and Fe_{1-x}Mn_xS. BET values ideally will corroborate particle size estimates from PXRD, but will also provide distribution information that could be pertinent. This is essential, because in catalysis (from Fischer-Tropsch to RWGS reactions) the preferred mechanisms, favouring of competitive pathways (e.g. carbon path vs hydrogen path in RWGS reactions), and concentration of active reaction intermediates all very strongly depend on reaction temperature, support composition (in this case silica), and active catalytic species' particle size. If the size distributions are similar between FeS and Fe_{1-x}Mn_xS then the comparison of catalytic activity is fair.

Response: Thank you for your insightful comment regarding the need for BET active surface area measurements in our study. In the revised manuscript, we have included BET surface area

analyses. Specifically, the FeS and Fe(Mn)S catalysts show very similar specific surface areas, with only minor differences between the two, indicating that the Mn indeed serves as a chemically reactive site for promoting catalytic behavior. By confirming that the surface area is similar between FeS and Fe(Mn)S, we can ensure a fair comparison of their catalytic activities.

Action: In the revised supplementary information page 6:

Table S1. Surface area and crystalline size of FeS and Fe(Mn)S catalysts.

Sample name	Specific surface area (m ² g ⁻¹) ^a	Crystalline size (nm) ^b
FeS	167.8	12.2
Fe(Mn)S	166.4	13.0

^a analyzed by Brunauer-Emmett-Teller (BET) method on the basis of nitrogen adsorption/desorption isotherms

^b estimated by the Scherrer equation based on the X-ray diffraction (XRD) analyses

7. The discussion of band gap analysis needs refinement, comparison to known literature values, and improved comparison to the authors' own DFT results. How do your measurements of band gap compare to expected scaling by Vegard's law, in a compound of similar size and charge species and capable of solid solution formation, the compound's unit cell parameters shift linearly. This will be important because the band gap of FeS is smaller than MnS in pure materials; it is expected that the substitution of MnS into FeS would result in an increase in band gap (i.e. if Fe were added to MnS band gap would decrease). You are reporting the opposite with strong data, so why is this? Is the decrease in bandgap a result of carrier concentrations increasing or defect site increases that affect the band gap energy or direct vs. indirect nature of it? Some reference data and optical results are found in Tigwere, G. A., et al. (2023; Materials Science in Semiconductor Processing, 158, 107365) for Fe incorporation into MnS, but the reverse arguments can be made quite strongly too. Similar style of combined experimental-computational bandgap analysis is done by Qin et al. in the study of MnS/BaS catalysts for instance (ACS Appl. Mater. Interfaces 2024, 16, 16, 20463–20471).

Response: Vegard's law predicts that in a solid solution, the unit cell parameters—and consequently, the band gap—should shift linearly with the composition of the material. However, such a tendency is not always captured fully by simplistic linear models like Vegard's law. For instance, Kong et al. (2019) created a series of BiOBr_xI_{1-x} compounds, and they found that the bandgap changes didn't follow Vegard's law because replacing bromine with iodine simultaneously

modulates the edges of the conduction and valence bands of the BiOBr, leading to nonlinear dependence of bandgap on the halogen anion concentrations.

Theoretically, incorporating metal ions into the crystal lattice of a host semiconductor introduces impurity levels within the forbidden band, which typically leads to a broadening of the optical response and a narrowing of the band gap (Di et al., 2018). This phenomenon occurs because the impurity levels create additional energy states that electrons can occupy, thereby reducing the energy required for electronic transitions.

In our study, we observed a slight decrease in the band gap upon the introduction of Mn into FeS. The incorporation of Mn likely introduces impurity states within the band structure, which can facilitate electron transitions at lower energy levels, reducing the band gap. This aligns well with previous reported values and trends (Yaqoob et al., 2022). The narrowing of the band gap might be attributed to the interaction between electrons and the introduced impurity levels, as well as the increase in charge carrier concentration caused by Mn doping (Yaqoob et al., 2022). In our manuscript, DFT calculations were specifically used to investigate the thermodynamic properties of each reaction step during CO₂ hydrogenation over pure and Mn-doped FeS, rather than to analyze changes in the band gap structure.

Action: In the revised manuscript, we have included the discussion about changes of the band gap due to Mn doping. Please see **Line 116**: *“This slight decrease in the band gap observed upon introducing Mn into FeS is likely a result of Mn introducing impurity states within the band structure which facilitate electron transitions at lower energy levels and thereby reduce the band gap.”*

8. In the discussion: when referencing the Hudson et al. 2020 PNAS study, it should be referred to as Hudson and coworkers (as it was Hudson’s lab that led the study).

Thank you for pointing that out. We have revised the manuscript to correctly reference the Hudson et al. 2020 PNAS study as "Hudson and coworkers" to accurately reflect the contribution of Hudson's lab in leading the study.

9. The citations are mostly complete but several recent prebiotic chemistry references about iron-sulfide driven CO₂ reduction are missing; these should also be mentioned as to how this work compares to their results:

- Santos-Carballal David, Roldan Alberto, Dzade Nelson Y. and de Leeuw Nora H. 2018; Reactivity of CO₂ on the surfaces of magnetite (Fe₃O₄), greigite (Fe₃S₄) and mackinawite (FeS) Phil. Trans. R.

Soc. A.37620170065

- Roldan et al., Bio-inspired CO₂ conversion by iron sulfide catalysts under sustainable conditions. Chem. Commun., 2015, 51, 7501-7504
- Yamaguchi et al. 2014, Electrochemical CO₂ Reduction by Ni-containing Iron Sulfides: How Is CO₂ Electrochemically Reduced at Bisulfide-Bearing Deep-sea Hydrothermal Precipitates? Electrochimica Acta 141 311-318

Response: We thank the reviewer for this comment. In the revised manuscript, we supplemented the discussion and comparison with the references suggested by the reviewer.

Please see **Line 255** in the revised manuscript: “*Pioneering studies have demonstrated that the surface of iron sulfide compounds, such as FeS, exhibits a unique capacity for CO₂ adsorption and activation, while Fe₃S₄ has been shown to catalyze the conversion of CO₂ into small organic molecules, such as formic acid and methanol.*”

Reviewer #3 (Remarks to the Author):

Nan et al. present a set of experimental results exploring the catalysis of FeS minerals towards gaseous CO₂ reduction, under analogous terrestrial hot springs environments on early Earth. The experiments include pure and Ti/Ni/Mn/Co-doped FeS minerals, and also consider the influence of temperature and UV light. Results seem to reasonably demonstrate the potential of FeS-catalyzed carbon fixation to methanol, and support the origins of life in terrestrial hot springs. This study provides a pathway of prebiotic molecule synthesis before the presence of photosynthesis, which might be helpful to understand the origins of early life.

Response: We thank the reviewer for the positive comments.

Major comments:

1. As another important iron sulfide phase, pyrite (FeS₂) seems to be more thermodynamically stable than FeS, thus might be more abundant on early Earth's surface, what role pyrite minerals play in catalyzing carbon fixation in terrestrial hot springs?

Response: Thank you for this comment. Pyrite was proposed by Wächtershäuser (1990) as a key driver in early metabolic cycles. He suggested that the oxidation of FeS to pyrite could be coupled with the reduction of CO₂ by H₂, providing the energy needed to overcome the high energy barriers associated with CO₂ reduction and thus facilitating the synthesis of organic molecules in the prebiotic environment. However, this CO₂ reduction pathway has never been successfully demonstrated in practice (White et al., 2015; Camprubi et al., 2017). Furthermore, while some studies have shown that pyrite can facilitate the formation of simple organic molecules, these studies often start with more reactive substrates like CO rather than CO₂ (Huber and Wächtershäuser, 1998), while CO is thought to be much less abundant in the early atmosphere. In summary, while the idea of pyrite driving early metabolic cycles is intriguing, it faces significant practical and theoretical challenges, particularly regarding the efficiency and feasibility in prebiotic conditions.

2. This study only showed the first ~6 mins results of H₂-dependent CO₂ reduction experiments in the Supplementary Information (Figure S5 and S6), but geological evolution might not stop in a very

short time, I am curious about how the reactions go on longer timescales (e.g., > 10 mins, hours, days...)?

Response: To clarify, the ~6 minutes mentioned in our study refers to the gas chromatography (GC) residence time, not the total duration of the H₂-dependent CO₂ reduction experiments. The experiments themselves continue beyond this initial period (~2 hours for each experiment in this study), as geological processes indeed occur over much longer timescales. Indeed, understanding chemical reactions over “deep time” is a fundamental challenge to the entire field of prebiotic chemistry and beyond the scope of our present manuscript.

3. Different hot springs might be characterized by different temperatures, pH, and alkalinity, how the latter two environmental factors influence prebiotic CO₂ fixation, and how other potential metals (such as Ag, Zn, Au, Pt?) catalyze these reductions in terrestrial hot springs?

Response: Thank you for this comment. We agree that ancient hot springs likely exhibited varying temperatures, which is why we conducted experiments across a range of temperatures from 80 to 120 °C. We acknowledge that factors such as pH and temperature as well as different doping metals could end up affecting the catalyst produced in the aqueous phase of these hot springs. The metals included in this study (Ti, Ni, Mn, and Co) were selected because they are commonly found in hot springs. Investigating the effects of other metals and pH during FeS synthesis will be an important direction for our future research.

Minor comments:

Line 73. I would suggest changing “have remained” to “remain”
In the revised manuscript, we have made this change.

Line 96. Maybe change “following” to “then”
In the revised manuscript, we have made this change.

Line 106. Might be better to change “for” to “of”
In the revised manuscript, we have made this change.

Line 109. Add “spectroscopy” after “X-ray”
In the revised manuscript, we have made this change.

Line 111. Supplementary Figure S1 and S2 are not involved in the main text?

In the revised manuscript, we have checked all the supplementary figures and tables are cited in the main text.

Line 165. I would suggest changing “for” to “at”?

In the revised manuscript, we have made this change.

Line 498. Add a “for” between “water” and “a minimum”

In the revised manuscript, we have made this change.

Line 575. Delete one “atoms”

In the revised manuscript, we have made this change.

Reviewer #4 (Remarks to the Author):

Response: We thank the reviewer for the valuable feedbacks.

References

- Bada, J. L. (2013). New insights into prebiotic chemistry from Stanley Miller's spark discharge experiments. *Chemical Society Reviews*, 42(5), 2186-2196.
- Bada, J. L. (2023). Volcanic Island lightning prebiotic chemistry and the origin of life in the early Hadean eon. *Nature Communications*, 14(1), 2011.
- Camprubi, E., Jordan, S. F., Vasiliadou, R., & Lane, N. (2017). Iron catalysis at the origin of life. *IUBMB life*, 69(6), 373-381.
- Deamer, D. (2011). *First life: Discovering the connections between stars, cells, and how life began*. Univ of California Press.
- Di, J., Xiong, J., Li, H., & Liu, Z. (2018). Ultrathin 2D photocatalysts: electronic-structure tailoring, hybridization, and applications. *Advanced Materials*, 30(1), 1704548.
- Ferretti, A. M., Mondini, S., & Ponti, A. (2016). Manganese sulfide (MnS) nanocrystals: synthesis, properties, and applications. *Advances in Colloid Science*, 121-123.
- Gaier, J., Street, K., & Gustafson, R. (2010, August). Measurement of the solar absorptance and the thermal emittance of lunar simulants. In *40th International Conference on Environmental Systems* (p. 6025).
- Huber, C., & Wächtershäuser, G. (1998). Peptides by activation of amino acids with CO on (Ni, Fe) S surfaces: implications for the origin of life. *Science*, 281(5377), 670-672.
- Hudson, R., de Graaf, R., Strandoo Rodin, M., Ohno, A., Lane, N., McGlynn, S. E., ... & Sojo, V. (2020). CO₂ reduction driven by a pH gradient. *Proceedings of the National Academy of Sciences*, 117(37), 22873-22879.
- Kong, L., Guo, J., Makepeace, J. W., Xiao, T., Greer, H. F., Zhou, W., ... & Edwards, P. P. (2019). Rapid synthesis of BiOBr_xI_{1-x} photocatalysts: Insights to the visible-light photocatalytic activity and strong deviation from Vegard's law. *Catalysis Today*, 335, 477-484.
- Lane, N., & Xavier, J. C. (2024). To unravel the origin of life, treat findings as pieces of a bigger puzzle. *Nature*, 626(8001), 948-951.
- Li, H., Zhang, L., & Cao, Y. (2021). Synthesis of palladium-modified MnS photocatalysts with enhanced photocatalytic activity in the photoreduction of CO₂ to CH₄. *Applied Surface Science*, 541, 148519.
- Li, Y., Li, Y., Liu, Y., Wu, Y., Wu, J., Wang, B., ... & Lu, A. (2020). Photoreduction of inorganic carbon (+ IV) by elemental sulfur: Implications for prebiotic synthesis in terrestrial hot springs. *Science Advances*, 6(47), eabc3687.
- Lu, W. N., Luo, S., Zhao, Y., Xu, J., Yang, G., Picheau, E., ... & Ye, J. (2024). Bifunctional Co active site on dilute CoCu plasmonic alloy for light-driven H₂ production from methanol and water.

- Applied Catalysis B: Environmental, 343, 123520.
- Luo, S., Song, H., Ichihara, F., Oshikiri, M., Lu, W., Tang, D. M., ... & Ye, J. (2023). Light-induced dynamic restructuring of Cu active sites on TiO₂ for low-temperature H₂ production from methanol and water. *Journal of the American Chemical Society*, 145(37), 20530-20538.
- McNutt, S. R., & Thomas, R. J. (2015). Volcanic lightning. In *The encyclopedia of volcanoes* (pp. 1059-1067). Academic Press.
- Nebel, O., Rapp, R. P., & Yaxley, G. M. (2014). The role of detrital zircons in Hadean crustal research. *Lithos*, 190, 313-327.
- Preiner, M., Igarashi, K., Muchowska, K. B., Yu, M., Varma, S. J., Kleinermanns, K., ... & Martin, W. F. (2020). A hydrogen-dependent geochemical analogue of primordial carbon and energy metabolism. *Nature Ecology & Evolution*, 4(4), 534-542.
- Sojo, V., Herschy, B., Whicher, A., Camprubi, E., & Lane, N. (2016). The origin of life in alkaline hydrothermal vents. *Astrobiology*, 16(2), 181-197.
- Sun, Z., Ma, T., Tao, H., Fan, Q., & Han, B. (2017). Fundamentals and challenges of electrochemical CO₂ reduction using two-dimensional materials. *Chem*, 3(4), 560-587.
- Van Kranendonk, M. J., Deamer, D. W., & Djokic, T. (2017). Life springs. *Scientific American*, 317(2), 28-35.
- Wächtershäuser, G. (1990). Evolution of the first metabolic cycles. *Proceedings of the National Academy of Sciences*, 87(1), 200-204.
- White, L. M., Bhartia, R., Stucky, G. D., Kanik, I., & Russell, M. J. (2015). Mackinawite and greigite in ancient alkaline hydrothermal chimneys: identifying potential key catalysts for emergent life. *Earth and Planetary Science Letters*, 430, 105-114.
- Yaqoob, S., ul Hasan, N., Khalid, S., & Akhtar, M. S. (2022). Structural, morphological and optical study of manganese doped FeS (Mackinawite) nanostructures by chemical bath deposition (CBD) technique. *Journal of Material Science and Technology Research*, 9(1), 24-33.
- Sojo, V., Herschy, B., Whicher, A., Camprubi, E., & Lane, N. (2016). The origin of life in alkaline hydrothermal vents. *Astrobiology*, 16(2), 181-197.
- Zgonnik, V. (2020). The occurrence and geoscience of natural hydrogen: A comprehensive review. *Earth-Science Reviews*, 203, 103140.